# Online Pricing for Multi-User Multi-Item Markets

Yigit Efe Erginbas[1]    Thomas Courtade[1]    Kannan Ramchandran[1]    Soham Phade[2]

[1]University of California, Berkeley    [2]Wayve Technologies

## Abstract

Online pricing has been the focus of extensive research in recent years, particularly in the context of selling an item to sequentially arriving users. However, what if a provider wants to maximize revenue by selling multiple items to multiple users in each round? This presents a complex problem, as the provider must intelligently offer the items to those users who value them the most without exceeding their highest acceptable prices. In this study, we tackle this challenge by designing online algorithms that can efficiently offer and price items while learning user valuations from accept/reject feedback. We focus on three user valuation models (fixed valuations, random experiences, and random valuations) and provide algorithms with nearly-optimal revenue regret guarantees. In particular, for any market setting with $N$ users, $M$ items, and load $L$ (which roughly corresponds to the maximum number of simultaneous allocations possible), our algorithms achieve regret of order $O(NM \log \log(LT))$ under fixed valuations model, $\widetilde{O}(\sqrt{NMLT})$ under random experiences model and $\widetilde{O}(\sqrt{NMLT})$ under random valuations model in $T$ rounds.

## 1 Introduction

The ability to design algorithms that can achieve the optimal sale of goods to multiple users having time-varying valuations for each of the goods is both timely and relevant, given the explosion in the use of data in large-scale systems. This problem is commonly encountered in various contexts, such as in the e-commerce (Amazon, eBay), ride-share (Uber, Lyft), airline, and hotel (Airbnb, Booking.com) industries. Since the provider's goal of maximizing revenue can only be achieved through a delicate balance of considering prior transactions and adjusting offers and prices, it presents a unique opportunity to advance our understanding of dynamic pricing.

We presently consider the problem of designing algorithms that aim to optimize the sale of multiple goods to multiple users having time-varying valuations over the course of repeated rounds. At each round, the provider offers each item to a user at a chosen price, and users decide whether or not to buy, by comparing the offered price to their private valuation for the good. The provider may decide on offers and prices based on outcomes of prior transactions, but each individual user accepts or rejects their offer based only on their valuation for the current round. The provider's goal is to maximize the revenue accumulated in multiple rounds by judiciously selecting the offers and associated prices.

The provider, who is endowed with multiple items at each round, repeatedly offers these items to multiple users at well-chosen prices. In response, the provider obtains feedback regarding the acceptance or rejection decisions of the users and receives revenue for the accepted items. The provider's goal is to maximize the accumulated revenue over time by making offers that respect the endowment constraints and user demands. In the process of identifying the best way of offering the items to target users, the provider encounters challenges regarding two separate aspects of the problem; such challenges are addressed in our work. As depicted in Figure 1, the first challenge is to learn user preferences from interactive feedback; the second is to find offers and prices that will result in maximal revenue.

37th Conference on Neural Information Processing Systems (NeurIPS 2023).

The challenge of *learning* arises from the fact that the provider does not have knowledge of the user valuations ahead of time and hence has to learn them while continually taking action. On the other hand, the challenge of *offering and pricing* stems from both of the facts that (a) available items are scarce and (b) user demands are limited. Therefore, to achieve better outcomes, the provider needs to make careful decisions on which items to offer to which user within the limits of these constraints. However, since the valuations of the users are unknown, the offering and pricing decisions involve a trade-off between learning individual customer preferences in order to increase long-term revenues and earning short-term revenues by leveraging the information acquired so far.

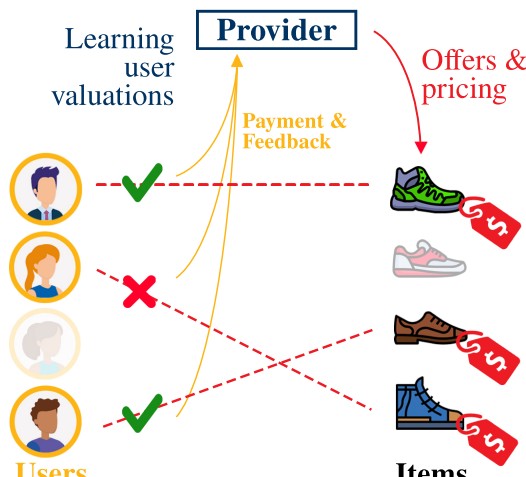

Figure 1: The provider's goal is to maximize the revenue obtained from sales over multiple rounds. The provider decides on offers and prices while interactively learning the user valuations from the accept/reject feedback. At each round, only a subset of items may be available for sale and only a subset of users may be active.

In this study, we focus on algorithms that offer each item to only one user during each round. This assumption eliminates the risk of multiple users requesting the same item and removes the need for the provider to decide who should receive it. Relaxing this assumption could possibly offer more flexibility and revenue potential during the earlier phases while learning the valuations. However, it does not result in a loss in the maximum achievable revenue under known valuations, and hence our algorithms can still achieve no-regret guarantees. Additionally, offering a limited number of items simplifies the process for users by reducing the number of choices they need to consider. As the provider gains insights into user valuations over time, the offers become more tailored and relevant, enabling users to focus on evaluating items that the provider believes will be of the highest interest to them. This saves time and reduces the effort required to make a decision.

## 1.1 Our contributions

To the best of our knowledge, we are the first to address the problem of dynamic pricing for the sale of multiple items to multiple users with unknown valuations. Our contributions are as follows.

- We consider three user valuation models: fixed valuations, random experiences, and random valuations. While the fixed valuation and random valuation models are standard models explored extensively in prior work on dynamic pricing (Bubeck et al., 2019), we also propose and analyze the random experiences model as a more realistic representation of user behavior.

- We introduce a problem-dependent load parameter $L$ that roughly corresponds to the maximum number of simultaneous allocations possible (see Def. 2). We uncover its crucial role in characterizing problem classes for which we can establish matching upper and lower regret bounds.

- We design regret-optimal (up to smaller order terms) algorithms for each setting. For any market setting with $N$ users, $M$ items, and maximum load $L$, our algorithms achieve regret $O(NM \log \log(LT))$ under fixed valuations model, $\tilde{O}(\sqrt{NMLT})$ under random experiences model and $\tilde{O}(\sqrt{NMLT})$ under random valuations model in $T$ rounds.

- All proposed algorithms have computational complexity $\mathcal{O}(NML)$ per round in the worst case. The algorithms for fixed valuations and random experiences models have space complexity $\mathcal{O}(NM)$ while the algorithm for random valuations has space complexity $\mathcal{O}(NM(LT)^{1/4})$.

## 2 Related work

**Bandits for dynamic posted-pricing.** The problem of dynamic pricing has been typically modeled as a variant of the multi-armed bandit problem starting with Rothschild (1974). In their seminal work, Kleinberg and Leighton (2003) developed a more general and widely-appreciated framework to maximize revenue from selling copies of a single good to sequentially arriving users. In the following years, there has been a growing body of work on multi-period single-product dynamic

pricing problems under different user valuation models including non-parametric models (Besbes and Zeevi, 2009; Keskin and Zeevi, 2014; Cesa-Bianchi et al., 2019; Bubeck et al., 2019), contextual (feature-based) models (Paes Leme and Schneider, 2018; Cohen et al., 2020; Xu and Wang, 2021), and other parametric models (Araman and Caldentey, 2009; Broder and Rusmevichientong, 2012; Harrison et al., 2012; Chen and Farias, 2013; Besbes and Zeevi, 2015; Ferreira et al., 2018). However, all of these works address the posted price problem for selling a single item in each round. Our contribution stands out by considering the combinatorial aspect of the allocation problem faced in *simultaneously selling multiple items*, a factor that was not taken into account in prior literature.

**Combinatorial multi-armed bandits.** The semi-bandits framework of Audibert et al. (2011) and the combinatorial multi-armed bandits frameworks of Chen et al. (2013) and Kveton et al. (2015) model problems where a player selects a combination of arms in each round and observes random rewards from the played arms. Therefore, the selection of the offers in our setting shows parallelism with these frameworks. However, their algorithmic solutions cannot be directly applied to our problem because the feedback and reward mechanisms in dynamic pricing are crucially different than the models considered in this literature. It is mainly because the feedback (acceptance/rejection) and reward (revenue) are not only affected by the offers but also by the accompanying prices.

**Bandits in matching markets.** One recent line of related literature in computational economics studies algorithms for learning socially-optimal matching in two-sided markets (Liu et al., 2020; Johari et al., 2017; Jagadeesan et al., 2021). These frameworks can be used to model the problem of allocating multiple items to multiple users with unknown valuations with the goal of maximizing social welfare. However, these works only consider scenarios where all the users accept their matchings (i.e. offers) without being affected by prices and send the provider real-valued random feedback representing the welfare they achieve from this matching. In their recent work, Erginbas et al. (2023) also analyze a similar problem of optimal and stable allocations and further allow users to accept or reject their recommendations based on the prices. However, their framework also requires random feedback to be sent to the provider regarding their valuation for each accepted matching, whereas our problem setting limits the provider to only observe acceptance or rejection decisions.

**Learning in repeated auctions.** The learning literature on auctions considers both offline (Morgenstern and Roughgarden, 2016; Cai and Daskalakis, 2017) and online approaches (Bar-Yossef et al., 2002; Lavi and Nisan, 2000) to maximize the provider's revenue from selling multiple items to multiple users. Nonetheless, the implementation of these auctions often presents significant difficulties due to the inherent complexities in obtaining precise valuations from human participants for all the items. As a result, there is a growing interest in designing mechanisms that are accessible, simple to use, and can easily elicit the valuations of the buyers while maximizing the revenue of the provider. In this direction, our approach in this study follows the main premise of posted-price mechanisms (Feldman et al., 2015; Einav et al., 2018), in which the provider sets a fixed price for each item, and buyers decide whether or not to purchase the item at that price.

## 3 Problem setting

**Notation:** We use bold font for vectors $\mathbf{x}$ and matrices $\mathbf{X}$, and calligraphic font $\mathcal{X}$ for sets. For a vector $\mathbf{x}$, we denote its $i$-th entry by $x_i$ and for a matrix $\mathbf{X}$, we denote its $(i,j)$-th entry by $x_{ij}$. For any positive integer $n$, we use $[n]$ to denote the set $\{1, 2, \ldots, n\}$. For real numbers $a$ and $b$, we use $a \wedge b$ to denote their minimum and $a \vee b$ to denote their maximum.

Suppose the market consists of a set of users $\mathcal{N}$ of size $N$ and a set of items $\mathcal{I}$ of size $M$. At each round $t \in [T]$ over some fixed time horizon $T$, a provider is endowed with a subset of the items denoted by $\mathcal{E}^t \subseteq \mathcal{I}$ and tries to sell these items to users with the goal of obtaining revenue. [1] We assume that endowed items $\mathcal{E}^t$ are only available for sale at time interval $t$. That is, the items that are not sold at a round cannot be stored to be sold in future rounds.

Each user $u \in \mathcal{N}$ has an *unknown* and possibly *time-varying* valuation $v_{ui}^t$ for each item $i$ at round $t$. In this work, we consider that each user $u$ has a time-varying request for at most $d_u^t$ items and has an additive utility model over items. Thus, the provider decides on a price vector $\boldsymbol{p}^t \in \mathbb{R}_+^M$ and offers each user $u \in \mathcal{N}$ a subset of the available items denoted by $\mathcal{S}_u^t \subseteq \mathcal{E}^t$ of size at most $|\mathcal{S}_u^t| \le d_u^t$. Then, users decide to accept a subset of their offered items $\mathcal{A}_u^t \subseteq \mathcal{S}_u^t$ based on their valuations and the prices of the items in order to maximize their surplus given by $\sum_{i \in \mathcal{A}_u^t} (v_{ui}^t - p_i^t)$.

---

[1]Our framework readily extends to scenarios where the provider is endowed with multiple copies of each type of item. However, for simplicity in our analysis and presentation, we do not consider this generalization.

Due to the additive utility assumption, when user $u$ is offered $\mathcal{S}_u^t$, it accepts all items $i \in \mathcal{S}_u^t$ that give positive surplus (i.e. $v_{ui}^t \geq p_i^t$) while rejecting all other items. [2] Hence, the set of accepted items can be written as

$$\mathcal{A}_u^t = \{i \in \mathcal{S}_u^t : v_{ui}^t \geq p_i^t\}.$$

For future reference, we also denote the collections of offered and accepted items at time $t$ by $\boldsymbol{\mathcal{S}}^t = \{\mathcal{S}_u^t | u \in \mathcal{N}\}$ and $\boldsymbol{\mathcal{A}}^t = \{\mathcal{A}_u^t | u \in \mathcal{N}\}$, respectively. To eliminate the possibility of an item getting accepted by multiple users, we consider posted-price offer mechanisms that offer each item to at most one user. Therefore, the offered sets of items are disjoint, i.e. $\mathcal{S}_u^t \cap \mathcal{S}_{u'}^t = \emptyset$ for $u \neq u'$.

Whenever a user accepts the offer of item $i$ at round $t$, this sale generates $p_i^t$ revenue for the provider. Therefore, the cumulative revenue obtained over $T$ rounds equals to

$$\sum_{t=1}^{T} \sum_{u \in \mathcal{N}} \sum_{i \in \mathcal{A}_u^t} p_i^t = \sum_{t=1}^{T} \sum_{u \in \mathcal{N}} \sum_{i \in \mathcal{S}_u^t} p_i^t \, \mathbb{1}\{v_{ui}^t \geq p_i^t\}. \tag{1}$$

After the provider decides on the price vector $\boldsymbol{p}^t$ and the offers $\boldsymbol{\mathcal{S}}^t = \{\mathcal{S}_u^t | u \in \mathcal{N}\}$, the users report their set of accepted items $\boldsymbol{\mathcal{A}}^t = \{\mathcal{A}_u^t | u \in \mathcal{N}\}$. We denote by $H_t$ the history $\{\boldsymbol{\mathcal{S}}^\tau, \boldsymbol{p}^\tau, \boldsymbol{\mathcal{A}}^\tau\}_{\tau=1}^{t-1}$ of observations available to the provider when choosing the next set of offers $\boldsymbol{\mathcal{S}}^t$ along with the next price vector $\boldsymbol{p}^t$. The provider employs a policy $\boldsymbol{\pi} = \{\pi^t | t \in \mathbb{N}\}$, which is a sequence of functions, each mapping the history $H_t$ to an action $(\boldsymbol{\mathcal{S}}^t, \boldsymbol{p}^t)$.

The task of the provider is to repeatedly offer the items to users and choose the prices such that it can achieve maximal revenue. To evaluate policies in achieving this objective, we define *regret* metrics that measure the gap between the expected revenue of policy $\boldsymbol{\pi}$ and an optimal algorithm.

**Definition 1.** *For a policy $\boldsymbol{\pi}$, its revenue regret in $T$ rounds is defined as*

$$\mathcal{R}(T, \boldsymbol{\pi}) = \text{OPT} - \sum_{t=1}^{T} \sum_{u \in \mathcal{N}} \sum_{i \in \mathcal{S}_u^t} p_i^t \, \mathbb{1}\{v_{ui}^t \geq p_i^t\}, \tag{2}$$

*where* OPT *denotes the revenue of the optimal algorithm which will be defined separately for each different valuation model in Section 4.*

We can also represent the offers $\boldsymbol{\mathcal{S}}^t = \{\mathcal{S}_u^t | u \in \mathcal{N}\}$ using binary variables $x_{ui}^t = \mathbb{1}\{i \in \mathcal{S}_u^t\}$ that indicate whether item $i$ is a member of each of the sets $\mathcal{S}_u^t$. With this definition, each variable $x_{ui}^t$ is equal to 1 if user $u$ is offered item $i$ at time $t$ and 0 otherwise. We collect these variables into a matrix $\boldsymbol{X}^t \in \{0,1\}^{N \times M}$ called the *offer* matrix. Due to endowment and demand constraints, the offer matrix $\boldsymbol{X}^t$ at each time $t$ needs to belong to the set

$$\mathcal{X}^t = \left\{ \boldsymbol{X} \in \{0,1\}^{N \times M} : \sum_{i \in \mathcal{I}} x_{ui}^t \leq d_u^t, \forall u \in \mathcal{N} \text{ and } \sum_{u \in \mathcal{N}} x_{ui} \leq e_i^t, \forall i \in \mathcal{I} \right\}, \tag{3}$$

where each endowment quantity $e_i^t = \mathbb{1}\{i \in \mathcal{E}^t\}$ is equal to 1 when item $i$ is available to be offered at time $t$, and 0 otherwise. Using this notation, we can write the cumulative revenue as

$$\sum_{t=1}^{T} \sum_{u \in \mathcal{N}} \sum_{i \in \mathcal{N}} x_{ui}^t p_i^t \, \mathbb{1}\{v_{ui}^t \geq p_i^t\}. \tag{4}$$

Lastly, we define the *maximum load* parameter which refers to the maximum amount of simultaneous demand and supply in the market. Formally,

**Definition 2.** *For a problem with endowment sequence $(\mathcal{E}^t)_{1 \leq t \leq T}$ and demand sequences $(d_u^t)_{1 \leq t \leq T}$ of all users $u \in \mathcal{N}$, the maximum load is defined as*

$$L = \max_{1 \leq t \leq T} \{D_t \wedge E_t\}, \tag{5}$$

*where $D_t := \sum_{u \in \mathcal{N}} d_u^t$ and $E_t := |\mathcal{E}^t|$ are the total demand and endowment at time $t$, respectively.*

Note that the maximum load parameter $L$ is an upper bound for the number of offers that can be made at any round $t$. This parameter is central to our analysis since the problem becomes more complex as the maximum load increases.

---

[2]In this definition, we assume that all tie-breaks are resolved in favor of the provider.

## 3.1 Summary of results

Our goal in this work is to provide insights into how to strategize multi-round posted-price offers when the provider does not have prior knowledge of user valuations. In particular, we consider the problem under three different valuation models as described below.

1. **Fixed valuations:** The valuations of users do not change over time. Formally, there exist values $v_{ui}$ such that $v_{ui}^t = v_{ui}$ for all $t \in [T]$.

2. **Random experiences:** The valuations of users are given as their average historical experience where each experience is independently drawn from distributions specific to each user and each item. Formally, we consider the experience of user $u$ with item $i$ to be given as the random variable $z_{ui}^t$ independently drawn from some distribution with cumulative distribution function $F_{ui}$ for all rounds at which user $u$ has accepted item $i$. Then, each valuation $v_{ui}^t$ is given as the average of the past experiences, i.e., the average of values $\{z_{ui}^\tau | \tau < t, i \in \mathcal{A}_u^t\}$.

3. **Random valuations:** The valuations of users at different rounds are independently drawn from distributions specific to each user and each item. Formally, there exist cumulative distribution functions $F_{ui}$ for all $(u, i) \in \mathcal{N} \times \mathcal{I}$ such that each $v_{ui}^t$ is independently drawn from a distribution with cdf $F_{ui}$.

For each of the models described above, we derive upper and lower bounds for revenue regret, matching up to logarithmic factors. We summarize our results in Table 1.

Table 1: Upper and lower bounds for revenue regret under different valuation models.

| Model | Upper Bound | Lower Bound |
|---|---|---|
| Fixed Valuations (Section 4.1) | $O\left(NM \log\log(LT)\right)$ | $\Omega\left(NM \log\log(LT/NM)\right)$ |
| Random Experiences (Section 4.2) | $\widetilde{O}\left(\sqrt{NMLT}\right)$ | $\Omega\left(\sqrt{NMLT}\right)$ |
| Random Valuations (Section 4.3) | $\widetilde{O}\left(\sqrt{NMLT}\right)$ | $\Omega\left(\sqrt{NMLT}\right)$ |

*Remark.* The frameworks of fixed valuations and random valuations can also be used to model settings where each interacting user is associated with a type that determines their valuations. In this case, the set $\mathcal{N}$ corresponds to the set of all user types and each demand parameter $d_u^t$ represents the total demand of users of type $u$ in round $t$. Under the fixed valuations model, all users of type $u$ have valuation $v_{ui}^t = v_{ui}$ for item $i$ at all rounds $t$. Under the random valuations model, each user of type $u$ has a random valuation with distribution $F_{ui}$ for item $i$ independently for each user at each time $t$. Since at most one user receives any item $i$ in any round, it is sufficient to consider a single random valuation $v_{ui}^t$ for each type $u$ and item $i$ at time $t$.

## 4 Methodology

We provide algorithms for achieving sub-linear revenue regret under different user valuation models described in Section 3.1. While the strategies for the selection of offers are similar under different models, we use different pricing strategies for different models as depicted in Figure 2.

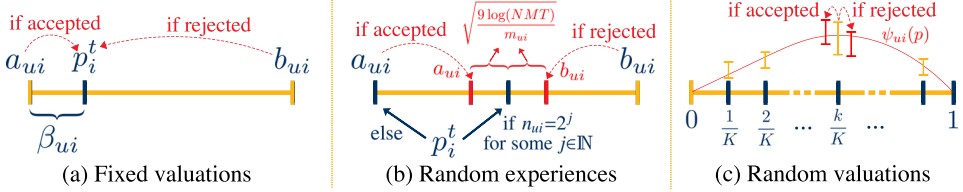

(a) Fixed valuations     (b) Random experiences     (c) Random valuations

Figure 2: The selection of prices for different valuation models. (a) If the offer is accepted (rejected), we set the new value of $a_{ui}$ ($b_{ui}$) as the offered price $p_i^t$. (b) If $n_{ui} = 2^j$ for some $j \in \mathbb{N}$ and the offer is accepted (rejected), we set the new value of $a_{ui}$ ($b_{ui}$) as a number smaller (larger) than $p_i^t$. (c) After offering an item at price $k/K$, we update the $k^{\text{th}}$ confidence interval based on whether the offer is accepted or rejected.

---

**Algorithm 1** Offerings with incremental search prices

---

$a_{ui} \leftarrow 0$, $b_{ui} \leftarrow 1$ and $\beta_{ui} \leftarrow 0.5$ for all $u \in \mathcal{N}, i \in \mathcal{I}$

**for** $t = 1, 2, \ldots, T$ **do**
    Calculate $\boldsymbol{X}^t$ by solving problem (8) using $b_{ui}$ values.
    **for** $(u, i) \in \{(u, i) : x_{ui}^t = 1\}$ **do**               ▷ offer item $i$ to user $u$
        $p_i^t \leftarrow a_{ui} + \beta_{ui} \mathbb{1}\{b_{ui} - a_{ui} \geq \frac{1}{LT}\}$              ▷ set the price
        Offer item $i$ to user $u$ at price $p_i^t$ and observe $\mathbb{1}\{v_{ui}^t \geq p_i^t\}$.
        **if** $v_{ui}^t \geq p_i^t$ **then** $a_{ui} \leftarrow a_{ui} \vee p_i^t$ **else** $b_{ui} \leftarrow b_{ui} \wedge p_i^t$      ▷ update the interval
        **if** $b_{ui} - a_{ui} \leq \beta_{ui}$ **then** $\beta_{ui} \leftarrow \beta_{ui}^2$          ▷ update the search step size
    **end for**
**end for**

---

## 4.1 Revenue maximization under fixed valuations

In this section, we focus on the scenario where the provider makes posted-price offers to users whose valuations are fixed over time. We formalize this condition in the following assumption.

**Assumption 1.** *The valuation of any user $u \in \mathcal{N}$ for any item $i \in \mathcal{I}$ is given by $v_{ui}^t = v_{ui}$ for some $v_{ui} \in [0, 1]$ at all rounds $t \in [T]$.*

In the case of fixed valuations, we define the optimum strategy as the one that maximizes the revenue under complete information on user valuations. Therefore, the optimum offers $\boldsymbol{X}_*^t$ and prices $\boldsymbol{p}_*^t$ for all rounds $t$ are given by maximizing the cumulative revenue as given in (4). That is, we define the optimum objective value as

$$\text{OPT} = \sum_{t=1}^{T} \left[ \max_{\boldsymbol{X}^t \in \mathcal{X}^t} \max_{\boldsymbol{p}^t \in \mathbb{R}_+^M} \sum_{u \in \mathcal{N}} \sum_{i \in \mathcal{I}} x_{ui}^t p_i^t \mathbb{1}\{v_{ui} \geq p_i^t\} \right], \tag{6}$$

and let $\boldsymbol{X}_*^t$ and $\boldsymbol{p}_*^t$ be the solutions that maximize term $t$. Note that for any allocation $\boldsymbol{X}^t \in \mathcal{X}^t$, we have $\sum_{u \in \mathcal{N}} x_{ui}^t \leq e_i^t \leq 1$. Therefore, whenever an item $i$ is offered to a user $u$, the optimum price of item $i$ is equal to the user's valuation $v_{ui}$. In other words, the optimum price for item $i$ is given by $p_i^t = \sum_{u \in \mathcal{N}} x_{ui}^t v_{ui}$. Based on this observation, the revenue-maximizing offer can be found as

$$\boldsymbol{X}_*^t = \arg\max_{\boldsymbol{X}^t \in \mathcal{X}^t} \sum_{u \in \mathcal{N}} \sum_{i \in \mathcal{I}} x_{ui}^t v_{ui}. \tag{7}$$

We note that the integer program in (7) can be written as an instance of maximum weight bipartite matching. Then, using a variant of the Hungarian algorithm for unbalanced bipartite graphs (Ramshaw and Tarjan, 2012), this problem can be solved in space $\mathcal{O}(NM)$ and time $\mathcal{O}(NML)$ in the worst case. (See Appendix A for details.)

Since there is no randomness in user responses, every response from a user $u$ about item $i$ gives the provider complete information about a lower or upper bound on $v_{ui}$, depending on whether the user response was to accept or reject the offered price for the item $i$. For this reason, the algorithm operates by keeping track of intervals $[a_{ui}, b_{ui}]$ that contain $v_{ui}$ value for each $u \in \mathcal{N}, i \in \mathcal{I}$ at all rounds. At each time $t$, the algorithm chooses which items to offer to each user using these intervals and then determines a price for each offered item. The selection of the allocations is done according to the OFU principle (Dani et al., 2008; Abbasi-Yadkori, 2011) in order to ensure low regret. In particular, the offers are chosen by replacing each $v_{ui}$ in problem (7) with $b_{ui}^t$ to obtain

$$\boldsymbol{X}^t = \arg\max_{\boldsymbol{X}^t \in \mathcal{X}^t} \sum_{u \in \mathcal{N}} \sum_{i \in \mathcal{I}} x_{ui}^t b_{ui}^t. \tag{8}$$

Having decided on the offers, the next step is to decide on the price of the offered items. Similar to the challenge we encountered in selecting the offers, the selection of prices should also serve two different goals that are in tension with each other. On one hand, learning new information about users' valuations requires us to set prices to values from $(a_{ui}, b_{ui})$. On the other hand, to ensure the offers are accepted and generate revenue, we would need to select prices lower than or equal to $a_{ui}$. The crucial property that enables us to obtain $O(NM \log \log(T))$ regret is that the function $p \rightarrow v_{ui} - p \mathbb{1}\{p \leq v_{ui}\}$ is asymmetric and decreases more slowly on the left than on the right as discussed in Kleinberg and Leighton (2003). Based on this observation, we design a pricing algorithm

that operates by offering item $i$ to user $u$ at prices increasing with increments of $\beta_{ui}$. When the width of the interval $[a_{ui}, b_{ui}]$ becomes smaller than the precision parameter $\beta_{ui}$, we set the new precision parameter as $\beta_{ui}^2$ and continue exploration with this smaller step size. However, as we show in the proof of Theorem 1, continuing to explore indefinitely would result in a regret linear in $T$. To avoid this issue, the algorithm should stop exploration after achieving a certain level of precision and offer item $i$ to user $u$ at the maximum price that is certainly acceptable, namely $a_{ui}$.

We provide a summary of this algorithm in Algorithm 1 and provide an upper bound for its regret in Theorem 1. As the lower bound provided in Theorem 2 shows, the regret of this algorithm is order-optimal up to smaller terms.

**Theorem 1** (Upper bound for Fixed valuations). *Assuming fixed valuations in a market with $N$ users, $M$ items, and maximum load $L$; the regret of Algorithm 1 in $T$ rounds satisfies*

$$\mathcal{R}(T, \boldsymbol{\pi}) \leq 2NM \log \log(LT) + 1.$$

*Proof.* See Appendix B.1. □

**Theorem 2** (Lower bound for Fixed valuations). *If $N$, $M$, and $L \leq M$ are given parameters and $\boldsymbol{\pi}$ is any randomized policy; there exist randomly generated market instances of $N$ users with fixed valuations for $M$ items and maximum load $L$ such that the expected regret of $\boldsymbol{\pi}$ in $T \geq \frac{4NM}{L}$ rounds is $\Omega(NM \log \log(LT/NM))$.*

*Proof.* See Appendix B.2. □

## 4.2 Revenue maximization under random experiences

This section focuses on the case where the valuations of the users are given as the average of their past experience. We assume that each user obtains independent experiences about the items that they accept at every round. Then, the users form their valuation at each time as the average of their experiences so far. Formally, we make the following assumption.

**Assumption 2.** *Whenever a user $u \in \mathcal{N}$ accepts an item $i \in \mathcal{I}$ at round $t \in [T]$, it obtains an independent random experience $z_{ui}^t$ with unknown cdf $F_{ui}$ over $[0, 1]$ with mean $v_{ui}$. Then, the valuation of any user $u \in \mathcal{N}$ for any item $i \in \mathcal{I}$ at round $t \in [T]$ is given as*

$$v_{ui}^t = \frac{1}{m_{ui}^t} \sum_{\tau \in \mathcal{T}_{ui}^t} z_{ui}^t, \tag{9}$$

*where $\mathcal{T}_{ui}^t = \{\tau < t : i \in \mathcal{A}_u^t\}$ is the set of rounds before round $t$ at which user $u$ has accepted and experienced item $i$, and $m_{ui}^t := |\mathcal{T}_{ui}^t|$ denotes the size of this set.*

Under the random experiences model, we define the optimum strategy as the one that maximizes the revenue with complete information on possible user experiences $z_{ui}^t$, and hence user valuations $v_{ui}^t$ at all times. Therefore, the optimum revenue that can be obtained in $T$ rounds is given by

$$\text{OPT} = \max_{\substack{\boldsymbol{X}^t \in \mathcal{X}^t : t \in [T] \\ \boldsymbol{p}^t \in \mathbb{R}_+^M : t \in [T]}} \sum_{t=1}^{T} \sum_{u \in \mathcal{N}} \sum_{i \in \mathcal{I}} x_{ui}^t p_i^t \mathbb{1}\{v_{ui}^t \geq p_i^t\}, \tag{10}$$

which is a random variable due to the randomness in $z_{ui}^t$ and hence the randomness in $v_{ui}^t$. Next, we note that for any fixed sequence of offers $\{\boldsymbol{X}^t | t \in [T]\}$, the objective is maximized at prices that satisfy $p_i^t = \sum_{u \in \mathcal{N}} x_{ui}^t v_{ui}$ at all $t$. Therefore, we can write the optimum value for the objective as

$$\text{OPT} = \max_{\boldsymbol{X}^t \in \mathcal{X}^t : t \in [T]} \sum_{t=1}^{T} \sum_{u \in \mathcal{N}} \sum_{i \in \mathcal{I}} x_{ui}^t v_{ui}^t. \tag{11}$$

Note that in problem (11), we need to globally maximize over $\boldsymbol{X}^t$ for all rounds $t \in [T]$ because the values of $v_{ui}^t$ depend on selections of $\boldsymbol{X}^t$ at previous rounds. In order to deal with this dependency between the selections of $\boldsymbol{X}^t$, in Lemma 9, we show that OPT is not likely to be much larger than the sum of the mean valuations of the best offers at all rounds.

---
**Algorithm 2** Offerings with scheduled learning
---
$a_{ui} \leftarrow 0$, $b_{ui} \leftarrow 1$, $n_{ui} \leftarrow 0$, and $m_{ui} \leftarrow 0$ for all $u \in \mathcal{N}, i \in \mathcal{I}$
**for** $t = 1, 2, \ldots, T$ **do**
    Calculate $\boldsymbol{X}^t$ by solving problem (8) using $b_{ui}$ values
    **for** $(u, i) \in \{(u, i) : x_{ui}^t = 1\}$ **do**                     ▷ offer item $i$ to user $u$
        **if** $\exists j \in \mathbb{N}$ s.t. $n_{ui} = 2^j$ **then** $p_i^t \leftarrow (a_{ui} + b_{ui})/2$ **else** $p_i^t \leftarrow a_{ui}$     ▷ set the price
        Offer item $i$ to user $u$ at price $p_i^t$ and observe $\mathbb{1}\{v_{ui}^t \geq p_i^t\}$.
        $\gamma \leftarrow \sqrt{8\log(NMT)/m_{ui}}$                         ▷ set the confidence level
        **if** $v_{ui}^t \geq p_i^t$ **then** $a_{ui} \leftarrow a_{ui} \vee (p_i^t - \gamma)$ **else** $b_{ui} \leftarrow b_{ui}^t \wedge (p_i^t + \gamma)$   ▷ update the interval
        **if** $v_{ui}^t \geq p_i^t$ **then** $m_{ui} \leftarrow m_{ui} + 1$
        $n_{ui} \leftarrow n_{ui} + 1$
    **end for**
**end for**
---

Next, we describe our algorithm in Algorithm 2. As the users accept and experience the items, their valuations converge to the mean of their experience distribution. Therefore, before extracting the mean valuation information from the users, the provider must ensure that users accept and sufficiently experience the items. Then, to extract this information, the algorithm occasionally asks for higher prices while taking the risk of rejected offers. However, since there are at most $\log(T)$ such learning rounds per user-item pair, it does not cause any significant loss in revenue.

Based on these observations, we establish an upper bound for our algorithm's regret in Theorem 3. Furthermore, as indicated by the regret lower bound presented in Theorem 4, our algorithm's regret is order-optimal up to smaller terms.

**Theorem 3** (Upper bound for random experiences). *In any market of $N$ users satisfying the random experience model given in Assumption 2 for $M$ items and maximum load $L$, with probability $1 - 2\delta$, the revenue regret of Algorithm 2 satisfies*

$$\mathcal{R}(T, \pi) = O\left(\sqrt{NMLT\log(NM/\delta)} + NM\log T\right).$$

*Proof.* See Appendix C.2.          □

**Theorem 4** (Lower bound for random experiences). *If $N$, $M$, and $L \leq M$ are given parameters and $\pi$ is any randomized policy; there exist randomly generated market instances of $N$ users satisfying the random experience model given in Assumption 2 for $M$ items and maximum load $L$ such that the expected regret of $\pi$ in $T \geq N$ rounds is $\Omega(\sqrt{NMLT})$.*

*Proof.* See Appendix C.3.          □

### 4.3 Revenue maximization under random valuations

In this section, we consider the case where the valuations of the users are given as independent random variables drawn from distributions specific to each user and item. Formally,

**Assumption 3.** *The valuation of any user $u \in \mathcal{N}$ for any item $i \in \mathcal{I}$ at round $t \in [T]$ is given as an independent random variable $v_{ui}^t$ with unknown cdf $F_{ui}$ over $[0, 1]$.*

Given foreknowledge of the distribution of valuations, but not of the individual valuations at different rounds, it is easy to see what the optimal pricing strategy would be. The expected revenue obtained from offering item $i$ to user $u$ at price $p$ is given as $\psi_{ui}(p) = p(1 - F_{ui}(p))$, which we call to be the revenue function. Since valuations are independent over time and their distribution is known, the individual responses provide no useful information about future realizations of the valuations. Therefore, the best price for offering item $i$ to user $u$ is given by

$$p_{ui}^* = \arg\max_{p \in \mathbb{R}_+} p(1 - F_{ui}(p)). \tag{12}$$

Thus, letting $\psi_{ui}^* = \psi_{ui}(p_{ui}^*)$ denote the maximum expected revenue that can be obtained by offering item $i$ to user $u$, an optimum policy that knows the distribution of valuations can obtain revenue

$$\text{OPT} = \sum_{t=1}^{T} \left\{ \max_{\boldsymbol{X} \in \mathcal{X}^t} \sum_{u \in \mathcal{N}} \sum_{i \in \mathcal{I}} x_{ui} \psi_{ui}^* \right\}. \tag{13}$$

---

**Algorithm 3** Offerings with quantized pricing

---

$K = (LT / (NM \log(LT)))^{1/4}$, $n_{uik} \leftarrow 0$, and $\psi_{uik} \leftarrow 1$ for all $u \in \mathcal{N}$, $i \in \mathcal{I}$, $k \in [K]$
**for** $t = 1, 2, \ldots, T$ **do**
    **for** $(u, i) \in \mathcal{N} \times \mathcal{I}$ **do**
        $b_{uik} \leftarrow \left( \psi_{uik} + \sqrt{8 \log(NMKT)/n_{uik}} \right) \wedge 1, \forall k \in [K]$           ▷ compute UCB
        $k_{ui} \leftarrow \arg\max_k b_{uik}$ and $b_{ui} \leftarrow \max_k b_{uik}$           ▷ compute best price levels
    **end for**
    Calculate $\boldsymbol{X}^t$ by solving problem (8) using $b_{ui}$ values.
    **for** $(u, i) \in \{(u, i) : x_{ui}^t = 1\}$ **do**           ▷ make offers
        Offer item $i$ to user $u$ at price $p_{ui} = k_{ui}/K$ and observe $z_{ui} = \mathbb{1}\{v_{ui}^t \geq p_{ui}\}$
        $\psi_{uik} \leftarrow (n_{uik}\psi_{uik} + p_{ui}z_{ui})/(n_{uik} + 1)$ for $k = k_{ui}$
        $n_{uik} \leftarrow n_{uik} + 1$ for $k = k_{ui}$
    **end for**
**end for**

---

However, without the knowledge of the distributions, we are required to learn the expected revenue (i.e. rewards) at different prices via exploration. As previously done in the literature on pricing under random valuation models (Kleinberg and Leighton, 2003), we apply techniques from the literature on the multi-armed bandit problem to develop an algorithm with low regret guarantees. To do so, we quantize the set of possible prices by limiting the provider to strategies that only offer prices belonging to the set $\{1/K, 2/K, \ldots 1\}$ for suitably chosen $K$. This brings us into a setting where each offer of item $i$ to a user $u$ at price $k/K$ yields a revenue which is a random variable taking values in $[0, 1]$, whose distribution depends on $(u, i, k)$, but the rewards for a given action are i.i.d. across the rounds. Therefore, offering item $i$ to user $u$ and $k^{th}$ price level can be represented as pulling an arm that generates revenue with expectation $\psi_{uik} = \psi_{ui}(k/K)$. In total, the expected revenue of any offering and pricing is

$$\sum_{t=1}^{T} \sum_{u \in \mathcal{N}} \sum_{i \in \mathcal{I}} \sum_{k=1}^{K} x_{uik}^t \psi_{uik}, \tag{14}$$

where $x_{uik}^t$ are binary variables that denote whether user $u$ is offered item $i$ at $k^{th}$ price level at round $t$. Due to endowment and demand constraints, these variables at time $t$ must satisfy conditions (1) $\sum_{i \in \mathcal{I}} \sum_{k=1}^{K} x_{uik} \leq d_u^t$, (2) $\sum_{u \in \mathcal{N}} \sum_{k=1}^{K} x_{uik} \leq e_i^t$, and (3) $\sum_{k=1}^{K} x_{uik} \leq 1$ for all $u \in \mathcal{N}, i \in \mathcal{I}$.

In the literature on combinatorial multi-armed bandits, the standard regret bounds for UCB-based algorithms have an inverse dependency on the gap between the rewards of optimal and suboptimal arms (Kveton et al., 2015). To use similar techniques in proving regret bounds for our algorithm, we make the following hypothesis on the distributions of valuations, which translates directly into bounds on price sub-optimality gaps $\max_k \psi_{uik} - \psi_{uik}$.

**Assumption 4.** *The revenue function $\psi_{ui}(p)$ has a unique global maximum at $p_{ui}^* \in (0, 1)$, and $\psi_{ui}''(p_{ui}^*)$ is defined and strictly negative.*

Next, we show that our algorithm can attain the regret upper bound stated in Theorem 5. Moreover, as illustrated in Theorem 6, our algorithm's regret is order-optimal up to smaller terms.

**Theorem 5** (Upper bound for random valuations). *In any market of $N$ users satisfying the random valuation model given in Assumptions 3 and 4 for $M$ items and maximum load $L$, with probability $1 - \delta$, the revenue regret of Algorithm 3 satisfies*

$$\mathcal{R}(T, \boldsymbol{\pi}) = \mathcal{O}\left( \sqrt{NMLT \log(LT) \log(NMT/\delta)} \right).$$

*Proof.* See Appendix D.2.         □

**Theorem 6** (Lower bound for random valuations). *If $N$, $M$, and $L \leq M$ are given parameters and $\boldsymbol{\pi}$ is any randomized policy; there exist randomly generated market instances of $N$ users satisfying the random valuation model given in Assumptions 3 and 4 for $M$ items and maximum load $L$ such that the expected regret of $\boldsymbol{\pi}$ is $\Omega(\sqrt{NMLT})$ in $T$ rounds.*

*Proof.* See Appendix D.3.         □

# 5  Numerical experiments

In this section, we demonstrate the efficacy of our proposed algorithms through a numerical study. We provide our results in Figures 3 and 4. At each round $t \in [T]$, the provider is endowed with each item $i \in \mathcal{I}$ (i.e. $i \in \mathcal{E}^t$) independently with probability $0.5$. On the other hand, each user $u \in \mathcal{N}$ has a random demand $d_u^t$ with uniform probability over $\{0, 1, 2\}$. For the case of fixed valuations, we choose each $v_{ui}$ independently from $\text{Beta}(2, 2)$. For other two models, we set each $F_{ui}$ as the cdf of $\text{Beta}(\alpha_{ui}, \beta_{ui})$ where $\alpha_{ui}$ and $\beta_{ui}$ are uniformly and independently chosen over $[1, 5]$.

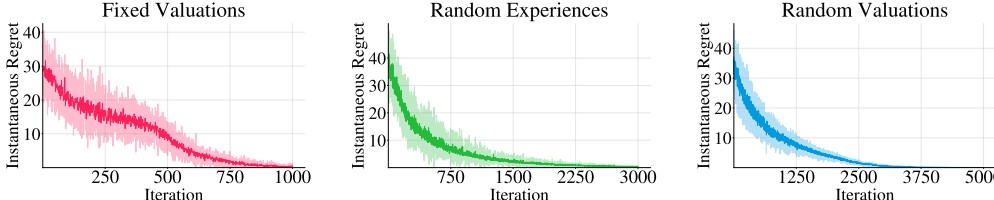

Figure 3: Instantaneous regret under different valuation models. The darker lines correspond to the mean across 20 experiments with $N = 150$ users and $M = 100$ items. The shaded areas indicate the region of error spanning two standard deviations. Results demonstrate the efficacy of our algorithms in achieving diminishing regret as our theoretical results predict.

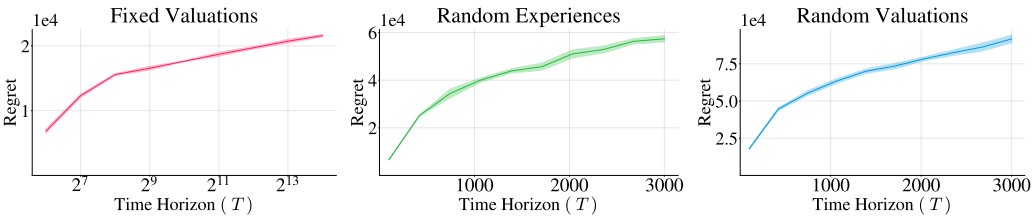

Figure 4: Regret as a function of time horizon $T$ under different valuation models. The darker lines correspond to the mean across 5 experiments with $N = 150$ users and $M = 100$ items. The shaded areas indicate the region of error spanning two standard deviations. Note that the horizontal axis is logarithmic in the plot for fixed valuations and linear in the plots for the other two cases. Results verify that our algorithms can achieve *sub-logarithmic regret* under the fixed valuations model, *sub-linear regret* under random experiences model, and *sub-linear regret* under random valuations model.

# 6  Conclusion

Our study presents a comprehensive solution to maximizing expected revenue in a repeated interaction setting, where a provider seeks to sell multiple items to multiple users. By focusing on different valuation models, we design online learning algorithms that can infer user valuations and offer items to those who value them most to ensure approximately optimal revenue regret. The results of this study have important implications for online marketplaces and can help providers optimize pricing strategies and maximize revenue in a dynamic and uncertain environment.

## Acknowledgements

This work was supported in part by NSF grants CIF-2007669 and CIF-1937357.

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

## A  Algorithmic Details for Solving the Integer Program

The integer linear program in (7) can be written as an instance of maximum weight bipartite matching. In this graph, we represent each user $u$ with $d_u^t$ left nodes and we represent each endowed item $i$ with a right node. Then, we construct the complete weighted bipartite graph where the weight of an edge between a node for user $u$ and a node for item $i$ is given as $v_{ui}$. In total, this graph has $D_t$ left vertices, $E_t$ right vertices, and $D_t E_t$ weighted edges where $D_t$ and $E_t$ correspond to total demand and endowment at time $t$ (as given in Definition 2).

Then, using a variant of the Hungarian algorithm for unbalanced bipartite graphs (Ramshaw and Tarjan, 2012), this problem can be solved in space $\mathcal{O}(D_t E_t)$ and time $\mathcal{O}(D_t E_t \min\{D_t, E_t\})$.

Since we can upper bound $D_t \leq N$, $E_t \leq M$, and $\min\{D_t, E_t\} \leq L$, we conclude that the algorithm runs in space $\mathcal{O}(NM)$ and time $\mathcal{O}(NML)$ in the worst case.

## B  Proofs for Revenue Maximization with Fixed Valuations

### B.1  Proof of Theorem 1

Letting $\mathrm{OPT}_t$ denote the optimum revenue at each time $t$ and $R_t$ denote the regret at round $t$,

$$\mathcal{R}_t = \mathrm{OPT}_t - \sum_{u \in \mathcal{N}} \sum_{i \in \mathcal{I}} x_{ui}^t p_i^t \mathbb{1}\{v_{ui} \geq p_i^t\}$$

$$= \max_{\boldsymbol{X} \in \mathcal{X}^t} \left\{ \sum_{u \in \mathcal{N}} \sum_{i \in \mathcal{I}} x_{ui} v_{ui} \right\} - \sum_{u \in \mathcal{N}} \sum_{i \in \mathcal{I}} x_{ui}^t p_i^t \mathbb{1}\{v_{ui} \geq p_i^t\}$$

$$\leq \max_{\boldsymbol{X} \in \mathcal{X}^t} \left\{ \sum_{u \in \mathcal{N}} \sum_{i \in \mathcal{I}} x_{ui} b_{ui}^t \right\} - \sum_{u \in \mathcal{N}} \sum_{i \in \mathcal{I}} x_{ui}^t a_{ui}^t \mathbb{1}\{v_{ui} \geq p_i^t\}$$

$$= \sum_{u \in \mathcal{N}} \sum_{i \in \mathcal{I}} x_{ui}^t b_{ui}^t - \sum_{u \in \mathcal{N}} \sum_{i \in \mathcal{I}} x_{ui}^t a_{ui}^t \mathbb{1}\{v_{ui} \geq p_i^t\}$$

$$= \sum_{u \in \mathcal{N}} \sum_{i \in \mathcal{I}} x_{ui}^t (b_{ui}^t - a_{ui}^t \mathbb{1}\{v_{ui} \geq p_i^t\}).$$

Then, we sum over all $1 \leq t \leq T$ to have

$$\mathcal{R}(T, \boldsymbol{\pi}) = \sum_{t=1}^T \mathcal{R}_t$$

$$\leq \sum_{t=1}^T \sum_{u \in \mathcal{N}} \sum_{i \in \mathcal{I}} x_{ui}^t (b_{ui}^t - a_{ui}^t \mathbb{1}\{v_{ui} \geq p_i^t\})$$

$$= \sum_{u \in \mathcal{N}} \sum_{i \in \mathcal{I}} \sum_{t \in \mathcal{T}_{ui}} (b_{ui}^t - a_{ui}^t \mathbb{1}\{v_{ui} \geq p_i^t\}),$$

where $\mathcal{T}_{ui} = \{t : 1 \leq t \leq T, x_{ui}^t = 1\}$ denotes the time indices where $(u, i)$ is offered. Now, we let

$$\bar{\mathcal{R}}_{ui} = \sum_{t \in \mathcal{T}_{ui}} (b_{ui}^t - a_{ui}^t \mathbb{1}\{v_{ui} \geq p_i^t\})$$

be an upper bound for the total regret incurred from offering the pair $(u, i)$ such that we have $\mathcal{R}(T, \boldsymbol{\pi}) \leq \sum_{u \in \mathcal{N}} \sum_{i \in \mathcal{I}} \bar{\mathcal{R}}_{ui}$. Now, let $k$th learning epoch for pair $(u, i)$ correspond to the time indices in which $\beta_{ui}^t = (1/2)^k$ and the pair $(u, i)$ is offered. That is,

$$\mathcal{T}_{ui}^k = \{t \in \mathcal{T}_{ui} : \beta_{ui}^t = (1/2)^k\}.$$

Note that we have $b_{ui}^t - a_{ui}^t \leq \beta_{ui}^t = (1/2)^k$ for each $t \in \mathcal{T}_{ui}^k$. During the learning phase, each epoch $\mathcal{T}_{ui}^k$ ends either when the offer $(u, i)$ is rejected or the offer $(u, i)$ is accepted $2^k - 1$ times in a row. Therefore, in epoch $k$, there are at most one rejection and $2^k - 1$ acceptances. As a result, the regret incurred by offers $(u, i)$ during each $\mathcal{T}_{ui}^k$ is upper bounded as

$$\sum_{t \in \mathcal{T}_{ui}^k} (b_{ui}^t - a_{ui}^t \mathbb{1}\{v_{ui} \geq p_i^t\}) \leq 1 + (2^k - 1)(1/2)^k \leq 2. \tag{15}$$

Note that a learning epoch $\mathcal{T}_{ui}^k$ can last for anywhere between 1 to $2^k$ rounds in total. Therefore, if we were to continue exploration indefinitely, the number of learning epochs could be as large as $T$ in the worst case. Consequently, continuing to explore indefinitely would result in linear regret. To avoid this issue, the algorithm should stop exploration after achieving a certain level of precision and offer item $i$ to user $u$ at the maximum price that is certainly acceptable, namely $a_{ui}$. We let this precision level be $\epsilon$ and make the algorithm choose price $a_{ui}$ when $b_{ui}^t - a_{ui}^t \leq \epsilon$. Based on this stopping rule, $K_{ui} = \lceil \log \log(1/\epsilon) \rceil$ becomes the last epoch for $(u, i)$ and therefore we run at most $K_{ui} - 1$ learning epochs before the last epoch.

Since the offers are always accepted in this last epoch, we incur at most $\epsilon$ regret at each round of it. Consequently, we have

$$\begin{aligned}
\overline{\mathcal{R}}_{ui} &\leq 2 \log \log(1/\epsilon) + \epsilon |\mathcal{T}_{ui}^{K_{ui}}| \\
&\leq 2 \log \log(1/\epsilon) + \epsilon |\mathcal{T}_{ui}|.
\end{aligned}$$

Since $\boldsymbol{X}^t$ includes at most $L$ offers, we also have $\sum_{u \in \mathcal{N}} \sum_{i \in \mathcal{I}} |\mathcal{T}_{ui}| \leq LT$. Thus,

$$\mathcal{R}(T, \boldsymbol{\pi}) \leq 2NM \log \log(1/\epsilon) + \epsilon LT. \tag{16}$$

Letting $\epsilon = 1/(LT)$, we obtain

$$\mathcal{R}(T, \boldsymbol{\pi}) \leq 2NM \log \log(LT) + 1. \tag{17}$$

### B.2 Proof of Theorem 2

Consider a market with users $\mathcal{N} = [N]$ where the demands of users are given as

$$d_u^t = \begin{cases} L, & \text{if} \quad t = (u-1) \mod N \\ 0, & \text{otherwise.} \end{cases}$$

so that only one user has nonzero demand in each round. Next, let $M'$ be the smallest prime number larger than or equal to $M$. If $N = M'$, let $M'$ be the next smallest prime number. By the prime number theorem, we always have $M' \leq 4M$, and $N$ and $M'$ are always co-prime. Then, consider a set of items $\mathcal{I} = [M']$ such that the valuation of all users for the first $M$ items are uniformly and independently chosen over $[0, 1]$ while the last $M' - M$ items are artificial items that have zero valuation for all users. Next, assume that the set of available items $\mathcal{E}^t$ at each time $t$ is given according to

$$i \in \mathcal{E}^t \quad \Longleftrightarrow \quad i - t + 1 \in \{1, 2, \ldots, L\} \mod M'$$

such that at most $L$ of the items are available at each round, and hence the maximum load parameter in this market is equal to $L$. By construction, the optimum offering pattern at each time is to offer all available items to a single user that has non-zero demand. Hence, the problem of the provider reduces to only learning the price at which it should offer each item. There are $NM$ actual user-item pairs and each item $i \in [M]$ is offered to user $u \in [N]$ for at least $\lfloor LT/NM' \rfloor \geq \lfloor LT/4NM \rfloor \geq 1$ rounds. In the literature on pricing optimization, each pricing problem is known to have $\Omega(\log \log(T_o))$ regret in $T_o$ rounds (Kleinberg and Leighton, 2003). Therefore, any policy must have $\Omega(NM \log \log(LT/NM))$ regret in total.

## C Proofs for Revenue Maximization with Random Experiences

### C.1 Preliminary Lemmas

**Lemma 7.** *With probability* $1 - \delta$,

$$|v_{ui}^t - v_{ui}| \leq \sqrt{\frac{8 \log(NM/\delta)}{m_{ui}^t}}$$

*for all* $(u, i) \in \mathcal{N} \times \mathcal{I}$ *and for all* $t \in \mathbb{N}$.

*Proof.* Let $\mathcal{H}_{t-1}$ be the $\sigma$-algebra generated by $(H_t, \boldsymbol{X}^t, \boldsymbol{p}^t)$ and let $\mathcal{H}_0 = \sigma(\emptyset, \Omega)$. Fix some $(u, i)$ and define $\epsilon_{ui}^t := z_{ui}^t - v_{ui}$ for all $t \in \mathbb{N}$. By previous assumptions, $\mathbb{E}[\epsilon_{ui}^t | \mathcal{H}_{t-1}] = 0$ and $\mathbb{E}[\exp(\lambda \epsilon_{ui}^t) | \mathcal{H}_{t-1}] \leq \exp(\lambda^2/2)$ for all $t \in \mathbb{N}$.

Define $\delta_{ui}^t := [(z_{ui}^t - v_{ui})^2 - (z_{ui}^t - v_{ui}^t)^2]\mathbb{1}\{x_{ui}^t = 1\}$. Then, we have

$$\delta_{ui}^t = \left[-(v_{ui}^t - v_{ui})^2 + 2\epsilon_{ui}^t(v_{ui}^t - v_{ui})\right]\mathbb{1}\{x_{ui}^t = 1\}.$$

Therefore, the conditional mean and conditional cumulant generating function satisfy

$$\mu_{ui}^t := \mathbb{E}[\delta_{ui}^t|\mathcal{H}_{t-1}] = -(v_{ui}^t - v_{ui})^2\mathbb{1}\{x_{ui}^t = 1\}$$
$$\psi_{ui}^t(\lambda) := \log\mathbb{E}[\exp(\lambda[\delta_{ui}^t - \mu_{ui}^t])|\mathcal{H}_{t-1}]$$
$$= \log\mathbb{E}[\exp(2\lambda\epsilon_{ui}^t(v_{ui}^t - v_{ui}))|\mathcal{H}_{t-1}]\mathbb{1}\{x_{ui}^t = 1\}$$
$$\leq \frac{(2\lambda(v_{ui}^t - v_{ui}))^2}{2}\mathbb{1}\{x_{ui}^t = 1\}.$$

Applying Lemma 14, we have, for all $x \geq 0$ and $\lambda \geq 0$,

$$\mathbb{P}\left(\sum_{\tau=1}^{t-1}\delta_{ui}^\tau \leq \frac{x}{\lambda} + \sum_{\tau=1}^{t-1}(v_{ui}^t - v_{ui})^2(2\lambda - 1)\mathbb{1}\{x_{ui}^\tau = 1\} \quad, \forall t \in \mathbb{N}\right) \geq 1 - e^{-x}.$$

Therefore,

$$\mathbb{P}\left(\sum_{\tau\in\mathcal{T}_{ui}^t}[(z_{ui}^\tau - v_{ui})^2 - (z_{ui}^\tau - v_{ui}^t)^2] \leq \frac{x}{\lambda} + \sum_{\tau\in\mathcal{T}_{ui}^t}(v_{ui}^t - v_{ui})^2(2\lambda - 1) \quad, \forall t \in \mathbb{N}\right) \geq 1 - e^{-x}.$$

Noting that $v_{ui}^t = \frac{1}{m_{ui}^t}\sum_{\tau\in\mathcal{T}_{ui}^t}z_{ui}^\tau$, we have $\sum_{\tau\in\mathcal{T}_{ui}^t}[(z_{ui}^\tau - v_{ui})^2 - (z_{ui}^\tau - v_{ui}^t)^2] \geq 0$ for all $t$. Then, choosing $\lambda = \frac{1}{4}$ and $x = \log\frac{1}{\delta}$ gives

$$\mathbb{P}\left(\sum_{\tau\in T_{ui}(t)}(v_{ui}^t - v_{ui})^2 \leq 8\log(1/\delta) \quad, \forall t \in \mathbb{N}\right) \geq 1 - \delta,$$

which implies

$$\mathbb{P}\left(|v_{ui}^t - v_{ui}| \leq \sqrt{\frac{8\log(1/\delta)}{m_{ui}^t}} \quad, \forall t \in \mathbb{N}\right) \geq 1 - \delta.$$

By applying a union bound over all $(u, i)$ pairs, we have

$$\mathbb{P}\left(|v_{ui}^t - v_{ui}| \leq \sqrt{\frac{8\log(NM/\delta)}{m_{ui}^t}} \quad, \forall(u, i) \in \mathcal{N} \times \mathcal{I}, \forall t \in \mathbb{N}\right) \geq 1 - \delta.$$

$\square$

**Lemma 8.** *With probability $1 - \delta$,*
$$v_{ui} \in [a_{ui}^t, b_{ui}^t]$$
*for all $(u, i) \in \mathcal{N} \times \mathcal{I}$ and for all $t \in \mathbb{N}$.*

*Proof.* From Lemma 7, we have $|v_{ui}^t - v_{ui}| \leq \sqrt{8\log(NM/\delta)/m_{ui}^t}$ for all $(u, i) \in \mathcal{N} \times \mathcal{I}$ and for all $t \in [T]$ with probability $1 - \delta$. Assume that this condition holds true.

We prove the statement of the lemma by induction. Assume $v_{ui} \in [a_{ui}^t, b_{ui}^t]$ as the inductive hypothesis. Then, if user $u$ accepts item $i$ at price $p_i^t$, then we have $v_{ui}^t \geq p_i^t$. Therefore, $p_i^t - v_{ui} \leq v_{ui}^t - v_{ui} \leq \sqrt{8\log(NM/\delta)/m_{ui}^t}$. So, $a_{ui}^{t+1} = a_{ui}^t \vee (p_i^t - \sqrt{8\log(NM/\delta)/m_{ui}^t}) \leq v_{ui}$. Similarly, we can also show $b_{ui}^{t+1} = b_{ui}^t \wedge (p_i^t + \sqrt{8\log(NM/\delta)/m_{ui}^t}) \geq v_{ui}$. Therefore, $v_{ui}^{t+1} \in [a_{ui}^{t+1}, b_{ui}^{t+1}]$.

$\square$

**Lemma 9.** *With probability $1 - \delta$, the value of* OPT *satisfies*

$$\text{OPT} - \sum_{t=1}^{T}\left\{\max_{\boldsymbol{X}\in\mathcal{X}^t}\sum_{u\in\mathcal{N}}\sum_{i\in\mathcal{I}}x_{ui}v_{ui}\right\} \leq O\left(\sqrt{NMLT\log(NM/\delta)}\right). \tag{18}$$

*Proof.* Let us define

$$\widetilde{\text{OPT}} = \sum_{t=1}^{T} \left\{ \max_{\boldsymbol{X} \in \mathcal{X}^t} \sum_{u \in \mathcal{N}} \sum_{i \in \mathcal{I}} x_{ui} v_{ui} \right\}. \tag{19}$$

Then, using the definition of OPT and $\widetilde{\text{OPT}}$ together with Lemma 7, we have

$$\text{OPT} = \max_{\boldsymbol{X}^t \in \mathcal{X}^t : t \in [T]} \sum_{t=1}^{T} \sum_{u \in \mathcal{N}} \sum_{i \in \mathcal{I}} x_{ui}^t v_{ui}^t$$

$$= \max_{\boldsymbol{X}^t \in \mathcal{X}^t : t \in [T]} \left\{ \sum_{t=1}^{T} \sum_{u \in \mathcal{N}} \sum_{i \in \mathcal{I}} x_{ui}^t v_{ui} + \sum_{t=1}^{T} \sum_{u \in \mathcal{N}} \sum_{i \in \mathcal{I}} x_{ui}^t (v_{ui}^t - v_{ui}) \right\}$$

$$\leq \max_{\boldsymbol{X}^t \in \mathcal{X}^t : t \in [T]} \left\{ \sum_{t=1}^{T} \sum_{u \in \mathcal{N}} \sum_{i \in \mathcal{I}} x_{ui}^t v_{ui} \right\} + \max_{\boldsymbol{X}^t \in \mathcal{X}^t : t \in [T]} \left\{ \sum_{t=1}^{T} \sum_{u \in \mathcal{N}} \sum_{i \in \mathcal{I}} x_{ui}^t (v_{ui}^t - v_{ui}) \right\}$$

$$= \widetilde{\text{OPT}} + \max_{\boldsymbol{X}^t \in \mathcal{X}^t : t \in [T]} \left\{ \sum_{t=1}^{T} \sum_{u \in \mathcal{N}} \sum_{i \in \mathcal{I}} x_{ui}^t (v_{ui}^t - v_{ui}) \right\}$$

$$\leq \widetilde{\text{OPT}} + \max_{\boldsymbol{X}^t \in \mathcal{X}^t : t \in [T]} \left\{ \sum_{t=1}^{T} \sum_{u \in \mathcal{N}} \sum_{i \in \mathcal{I}} x_{ui}^t \sqrt{\frac{8 \log(NM/\delta)}{m_{ui}^t}} \right\}$$

$$\leq \widetilde{\text{OPT}} + \sqrt{8 \log(NM/\delta)} \max_{\boldsymbol{X}^t \in \mathcal{X}^t : t \in [T]} \left\{ \sum_{t=1}^{T} \sum_{u \in \mathcal{N}} \sum_{i \in \mathcal{I}} x_{ui}^t \sqrt{1/m_{ui}^t} \right\}.$$

Recall that $n_{ui}^t$ counts the number of times user $u$ is offered item $i$ and $m_{ui}^t$ counts the number of corresponding acceptances. Since an item $i$ can be rejected only when $n_{ui}^t = 2^j$ for some $j \in \mathbb{N}$, we can write

$$\sum_{t=1}^{T} \sum_{u \in \mathcal{N}} \sum_{i \in \mathcal{I}} x_{ui}^t \sqrt{\frac{1}{m_{ui}^t}} = \sum_{u \in \mathcal{N}} \sum_{i \in \mathcal{I}} \sum_{t : x_{ui}^t = 1} \sqrt{\frac{1}{m_{ui}^t}}$$

$$\leq \sum_{u \in \mathcal{N}} \sum_{i \in \mathcal{I}} \left( \sum_{\substack{t : x_{ui}^t = 1 \\ \nexists j \in \mathbb{N} : n_{ui}^t = 2^j}} \sqrt{\frac{1}{m_{ui}^t}} + \sum_{\substack{t : x_{ui}^t = 1 \\ \exists j \in \mathbb{N} : n_{ui}^t = 2^j}} \sqrt{\frac{1}{m_{ui}^t}} \right)$$

$$\leq \sum_{u \in \mathcal{N}} \sum_{i \in \mathcal{I}} \left( \sum_{k=1}^{m_{ui}^T} \sqrt{\frac{1}{k}} + \log T \right)$$

$$\leq \sum_{u \in \mathcal{N}} \sum_{i \in \mathcal{I}} \left( 2\sqrt{m_{ui}^T} + \log T \right)$$

where we upper bound the first term inside the parentheses using $\sum_{k=1}^{n} \frac{1}{\sqrt{k}} \leq \int_0^n \frac{\mathrm{d}x}{\sqrt{x}} = 2\sqrt{n}$ and we upper bound the second term by noting that it can have at most $\log T$ terms. Also note that the total number of offers over all time intervals is upper bounded by $LT$, and therefore $\sum_{u \in \mathcal{N}} \sum_{i \in \mathcal{I}} m_{ui}^T \leq LT$. Then, using the Cauchy-Schwartz inequality, we can write

$$\sum_{t=1}^{T} \sum_{u \in \mathcal{N}} \sum_{i \in \mathcal{I}} x_{ui}^t \sqrt{\frac{1}{m_{ui}^t}} \leq 2\sqrt{NM \sum_{u \in \mathcal{N}} \sum_{i \in \mathcal{I}} m_{ui}^T} + NM \log T$$

$$\leq O(\sqrt{NMLT} + NM \log T).$$

Putting all together, we conclude the proof of the lemma.

$\square$

## C.2 Proof of Theorem 3

By Lemma 8, the confidence bounds include the mean, i.e., $v_{ui} \in [a_{ui}^t, b_{ui}^t]$ with probability $1 - \delta$. Assume that this condition holds.

Then, the offered items are accepted whenever there does not exist $j \in \mathbb{N}$ such that $n_{ui}^t = 2^j$. So, we have that $m_{ui}^t \geq n_{ui}^t/2$ because $n_{ui}^t$ counts the number of times item $i$ is offered to user $u$, and $m_{ui}^t$ counts the number of times item $i$ is accepted by user $u$.

Furthermore, we recall that intervals are only updated at time step $t$ if $x_{ui}^t = 1$ and $n_{ui}^t = 2^j$ for some $j \in \mathbb{N}$. Therefore, we can show an upper bound for the width of the intervals $[a_{ui}^t, b_{ui}^t]$. Define the width $g_{ui}^t = b_{ui}^t - a_{ui}^t$. When the update happens, the interval's size $g_{ui}^t$ becomes at most $g_{ui}^{t-1}/2 + \sqrt{8 \log(NM/\delta)/m_{ui}^t} \leq g_{ui}^{t-1}/2 + \sqrt{16 \log(NM/\delta)/n_{ui}^t}$. Therefore,

$$g_{ui}^t \leq \sum_{j=1}^{K} \left(\frac{1}{2}\right)^{K-j} \sqrt{\frac{16 \log(NM/\delta)}{2^{j+1}}},$$

where $K = \log_2 n_{ui}^t$ is the number of times the interval is updated before time $t$. Then,

$$g_{ui}^t \leq \frac{\sqrt{16 \log(NM/\delta)}}{2^K} \sum_{j=1}^{K} \left(\sqrt{2}\right)^{j-1}$$

$$\leq \frac{\sqrt{16 \log(NM/\delta)}}{2^K} (2^{K/2} - 1)(1 + \sqrt{2})$$

$$\leq \sqrt{\frac{144 \log(NM/\delta)}{2^K}}$$

$$\leq \sqrt{\frac{144 \log(NM/\delta)}{2^{\log n_{ui}^t - 1}}}$$

$$\leq \sqrt{\frac{288 \log(NM/\delta)}{n_{ui}^t}}.$$

Now, let $L_{ui}^t = \{x_{ui}^t = 1, \exists j \in \mathbb{N}, n_{ui}^t = 2^j\}$ denote the learning event for pair $(u, i)$ at round $t$. Also note that $x_{ui}^t = 1$ at some $t$, then $p_i^t = \mathbb{1}\{\neg L_{ui}^t\}a_{ui}^t + \mathbb{1}\{L_{ui}^t\}(a_{ui}^t + b_{ui}^t)/2$. Next, we recall the definition of $\widetilde{\text{OPT}}$ in (19) and let

$$\boldsymbol{Y}^t = \arg\max_{\boldsymbol{X} \in \mathcal{X}^t} \sum_{u \in \mathcal{N}} \sum_{i \in \mathcal{I}} x_{ui} v_{ui}, \tag{20}$$

with entries $y_{ui}^t$. Then, the regret with respect to $\widetilde{\text{OPT}}$ is given as the difference

$$\widetilde{\mathcal{R}}(T, \boldsymbol{\pi}) := \widetilde{\text{OPT}} - \sum_{t=1}^{T} \sum_{u \in \mathcal{N}} \sum_{i \in \mathcal{I}} x_{ui}^t p_i^t \mathbb{1}\{v_{ui}^t \geq p_i^t\}$$

$$= \sum_{t=1}^{T} \left\{ \sum_{u \in \mathcal{N}} \sum_{i \in \mathcal{I}} y_{ui}^t v_{ui} - \sum_{u \in \mathcal{N}} \sum_{i \in \mathcal{I}} x_{ui}^t p_i^t \mathbb{1}\{v_{ui}^t \geq p_i^t\} \right\}.$$

Recalling that the confidence bounds include the mean, i.e., $v_{ui} \in [a_{ui}^t, b_{ui}^t]$,

$$\widetilde{\mathcal{R}}(T, \boldsymbol{\pi}) \leq \sum_{t=1}^{T} \left\{ \sum_{u \in \mathcal{N}} \sum_{i \in \mathcal{I}} y_{ui}^t v_{ui} - \sum_{u \in \mathcal{N}} \sum_{i \in \mathcal{I}} x_{ui}^t a_{ui}^t \mathbb{1}\{\neg L_{ui}^t\} \right\}$$

$$\leq \sum_{t=1}^{T} \left\{ \sum_{u \in \mathcal{N}} \sum_{i \in \mathcal{I}} y_{ui}^t v_{ui} - \sum_{u \in \mathcal{N}} \sum_{i \in \mathcal{I}} x_{ui}^t a_{ui}^t + \sum_{u \in \mathcal{N}} \sum_{i \in \mathcal{I}} x_{ui}^t a_{ui}^t \mathbb{1}\{L_{ui}^t\} \right\}$$

$$\leq \sum_{t=1}^{T} \left\{ \sum_{u \in \mathcal{N}} \sum_{i \in \mathcal{I}} y_{ui}^t v_{ui} - \sum_{u \in \mathcal{N}} \sum_{i \in \mathcal{I}} x_{ui}^t a_{ui}^t \right\} + \sum_{t=1}^{T} \sum_{u \in \mathcal{N}} \sum_{i \in \mathcal{I}} \mathbb{1}\{L_{ui}^t\}.$$

Since $\boldsymbol{X}^t$ is selected according to

$$\boldsymbol{X}^t = \arg\max_{\boldsymbol{X} \in \mathcal{X}} \sum_{u \in \mathcal{N}} \sum_{i \in \mathcal{I}} x_{ui} b_{ui}^t, \tag{21}$$

we can upper bound the first term as

$$\begin{aligned}
\sum_{u \in \mathcal{N}} \sum_{i \in \mathcal{I}} (y_{ui}^t v_{ui} - x_{ui}^t a_{ui}^t) &\leq \sum_{u \in \mathcal{N}} \sum_{i \in \mathcal{I}} (y_{ui}^t b_{ui}^t - x_{ui}^t a_{ui}^t) \\
&\leq \sum_{u \in \mathcal{N}} \sum_{i \in \mathcal{I}} (x_{ui}^t b_{ui}^t - x_{ui}^t a_{ui}^t) \\
&= \sum_{u \in \mathcal{N}} \sum_{i \in \mathcal{I}} x_{ui}^t g_{ui}^t.
\end{aligned}$$

On the other hand, the second term is upper bounded as

$$\sum_{t=1}^{T} \sum_{u \in \mathcal{N}} \sum_{i \in \mathcal{I}} \mathbb{1}\{L_{ui}^t\} \leq NM \log T.$$

Therefore, we can upper bound $\widetilde{\mathcal{R}}(T, \boldsymbol{\pi})$ as

$$\widetilde{\mathcal{R}}(T, \boldsymbol{\pi}) \leq \sum_{t=1}^{T} \sum_{u \in \mathcal{N}} \sum_{i \in \mathcal{I}} x_{ui}^t g_{ui}^t + NM \log T.$$

Then, we can show

$$\begin{aligned}
\sum_{t=1}^{T} \sum_{u \in \mathcal{N}} \sum_{i \in \mathcal{I}} x_{ui}^t g_{ui}^t &\leq \sum_{t=1}^{T} \sum_{u \in \mathcal{N}} \sum_{i \in \mathcal{I}} x_{ui}^t \sqrt{\frac{288 \log(NM/\delta)}{n_{ui}^t}} \\
&= \sqrt{288 \log(NM/\delta)} \sum_{t=1}^{T} \sum_{u \in \mathcal{N}} \sum_{i \in \mathcal{I}} x_{ui}^t \sqrt{1/n_{ui}^t} \\
&\leq O(\sqrt{NMLT \log(NM/\delta)}),
\end{aligned}$$

where the proof of the last step is similar to the proof of Lemma 9. Therefore, with probability $1 - \delta$,

$$\widetilde{\mathcal{R}}(T, \boldsymbol{\pi}) \leq O(\sqrt{NMLT \log(NM/\delta)} + NM \log T).$$

Then, we use Lemma 9 which holds with probability $1 - \delta$. Therefore, by applying a union bound, we can show that

$$\begin{aligned}
\mathcal{R}(T, \pi) &= \mathrm{OPT} - \widetilde{\mathrm{OPT}} + \widetilde{\mathcal{R}}(T, \boldsymbol{\pi}) \\
&\leq O(\sqrt{NMLT \log(NM/\delta)} + NM \log T),
\end{aligned}$$

with probability $1 - 2\delta$.

### C.3  Proof of Theorem 4

Recall that our observations are limited to accept-reject signals given by indicator variables $\mathbb{1}\{v_{ui}^t \geq p_i\}$ for all $(u, i)$ such that $x_{ui}^t = 1$. Furthermore, the valuations are given by

$$v_{ui}^t = \frac{1}{n_{ui}^t} \sum_{\tau \in \mathcal{T}_{ui}^t} z_{ui}^t,$$

where $z_{ui}^t$ are i.i.d. random variables with mean $v_{ui}$ and $\mathcal{T}_{ui}(t) = \{\tau < t : x_{ui}^t = 1\}$. Hence, the signal $\mathbb{1}\{v_{ui}^t \geq p_i\}$ is a function of $z_{ui}^t$ in $\mathcal{T}_{ui}(t)$. As a result, the problem of learning revenue-maximizing offers by observing $z_{ui}^t$ for all $(u, i)$ such that $x_{ui}^t = 1$ is no harder than that of learning revenue-maximizing offers by observing $\mathbb{1}\{v_{ui}^t \geq p_i\}$ signals for all $(u, i)$ such that $x_{ui}^t = 1$.

On top of observing $z_{ui}^t$, if we further assume that an oracle sets the prices $p_i = v_{ui}^t$ whenever $x_{ui}^t = 1$ (such that all the offers are accepted by achieving revenue $v_{ui}^t$), the corresponding problem becomes an online linear optimization problem with semi-bandit feedback. In this problem, at each round $t$, the algorithm chooses an offering $X^t$ from the set of feasible offers $\mathcal{X}^t$, observes $z_{ui}^t$ for all $(u, i)$ such that $x_{ui}^t = 1$, and aims to maximize the total revenue (note that the revenue at time $t$ is equal to the sum of valuations for all $(u, i)$ such that $x_{ui}^t = 1$). Since this problem is also no harder than our original problem, any lower bound for this problem also applies to our setting.

Let $\mathcal{N} = [N]$ and $\mathcal{I} = [M]$ be the set of users and items respectively. Then, consider a market setting where the demands of users are given as

$$d_u^t = \begin{cases} L, & \text{if} \quad t = (u-1) \mod N \\ 0, & \text{otherwise} \end{cases}$$

so that only one user has nonzero demand in each round and each user is active for $\Theta(T/N)$ rounds. Let $\mathcal{E}_t = \mathcal{I}$ for all rounds and note that the maximum load in this market is $L$ as needed. Consequently, we have $N$ independent sub-problems where each sub-problem corresponds to offering the best $L$ items in $\mathcal{I}$ to a single user. In the literature on online linear optimization with semi-bandit feedback, this problem is known as the L-sets problem and it is known that for any algorithm there exists an instance such that expected regret $\Omega(\sqrt{LMT_o})$ in $T_o$ rounds Kleinberg and Leighton (2003). Therefore, for any algorithm, each of $N$ sub-problems has $\Omega(\sqrt{LMT/N})$ regret which corresponds to $\Omega(\sqrt{NMLT})$ regret in total for the whole problem.

# D  Proofs for Revenue Maximization with Random Valuations

## D.1  Preliminary Lemmas

**Lemma 10.** *For any $u \in \mathcal{N}$ and $i \in \mathcal{I}$, there exists constants $C_1$ and $C_2$ such that*

$$C_1 (p - p_{ui}^*)^2 < \psi_{ui}(p_{ui}^*) - \psi_{ui}(p) < C_2 (p - p_{ui}^*)^2 \tag{22}$$

*for all $p \in [0, 1]$.*

*Proof.* Since $\psi_{ui}''(p_{ui}^*)$ is strictly negative, there exists constants $A_1$, $A_2$, $\epsilon > 0$ such that $A_1(p - p_{ui}^*)^2 < \psi_{ui}(p_{ui}^*) - \psi_{ui}(p) < A_2(p - p_{ui}^*)^2$ for all $p \in (p_{ui}^* - \epsilon, p_{ui}^* + \epsilon)$. Since the set $\mathcal{S} = \{p \in [0, 1] : |p - p_{ui}^*| \geq \epsilon\}$ is compact and $\psi_{ui}(p_{ui}^*) - \psi_{ui}(p) > 0$ for all $p \in \mathcal{S}$, there exists constants $B_1$, $B_2$ such that $B_1(p - p_{ui}^*)^2 < \psi_{ui}(p_{ui}^*) - \psi_{ui}(p) < B_2(p - p_{ui}^*)^2$ for all $p \in \mathcal{S}$. Hence, if we set $C_1 = A_1 \wedge B_1$ and $C_2 = A_2 \vee B_2$, we obtain the statement of the lemma. $\square$

**Lemma 11.** *For any $u \in \mathcal{N}$ and $i \in \mathcal{I}$, define $\Delta_{uik} = \max_k \psi_{uik} - \psi_{uik}$. If $\widetilde{\Delta}_{ui0} \leq \widetilde{\Delta}_{ui1} \leq \cdots \leq \widetilde{\Delta}_{ui(K-1)}$ are the elements of the set $\{\Delta_{ui1}, \ldots, \Delta_{uiK}\}$ sorted in ascending order, then $\widetilde{\Delta}_{uik} \geq c_1(k/2K)^2$.*

*Proof.* Applying Lemma 10 and using the definition of $\Delta_{uik}$, we can show that $\Delta_{uik} \geq C_1(p_{ui}^* - k/K)^2$. Then, lower bound on $\widetilde{\Delta}_{uik}$ follows upon observing that at most $j$ elements of the set $\{1/K, 2/K, \ldots, 1\}$ lie within a distance $j/2K$ of $p_{ui}^*$. $\square$

**Lemma 12.** *For any $u \in \mathcal{N}$ and $i \in \mathcal{I}$, we have $\psi_{ui}^* - \max_k \psi_{uik} \leq C_2/K^2$.*

*Proof.* Applying Lemma 10 and noting that at least one of the prices $\{1/K, 2/K, \ldots, 1\}$ lies within a distance $1/K$ of $p_{ui}^*$, we obtain the statement of the lemma. $\square$

## D.2  Proof of Theorem 5

Let $\pi^*$ be the algorithm that chooses the optimum choice of offers and prices. Let $\pi_K^*$ be the algorithm that chooses the optimum choice of offers, but chooses the prices as the best price from the set $\{1/K, 2/K, \ldots, 1\}$. Let $\pi$ denote our algorithm described in Algorithm 3. Furthermore, let $\rho(\cdot)$ represent the expected revenue obtained by an algorithm. Then, the regret of policy $\pi$ satisfies

$$\mathcal{R}(T, \pi) = \rho(\pi^*) - \rho(\pi) = (\rho(\pi^*) - \rho(\pi_K^*)) + (\rho(\pi_K^*) - \rho(\pi)).$$

We begin with upper bounding $\rho(\boldsymbol{\pi}_K^*) - \rho(\boldsymbol{\pi})$. Recall that we can represent the expected revenue of any offering and pricing (that is constrained to set the set $\{1/K, 2/K, \ldots, 1\}$) as

$$\sum_{u \in \mathcal{N}} \sum_{i \in \mathcal{I}} \sum_{k=1}^{K} x_{uik}^t \psi_{uik},$$

where $x_{uik}^t$ are binary variables that denote whether user $u$ is offered item $i$ at $k^{th}$ price level. Due to the endowment, demand constraints, and the constraint that requires the prices to be from the set $\{1/K, 2/K, \ldots, 1\}$, the $x_{uik}^t$ variables at each time $t$ need to belong to the set

$$\mathcal{X}_e^t = \Big\{ \boldsymbol{X} \in \{0,1\}^{N \times M \times K} : \sum_{i \in \mathcal{I}} \sum_{k=1}^{K} x_{uik} \leq d_u^t, \forall u \in \mathcal{N}$$

$$\text{and } \sum_{u \in \mathcal{N}} \sum_{k=1}^{K} x_{uik} \leq e_i^t, \forall i \in \mathcal{I}$$

$$\text{and } \sum_{k=1}^{K} x_{uik} \leq 1, \forall u \in \mathcal{N}, i \in \mathcal{I} \Big\}.$$

Similar to the proof of Lemma 7, we have $|\widehat{\psi}_{uik}^t - \psi_{uik}| \leq \sqrt{8 \log(NMK/\delta)/n_{uik}^t}$ for all $u \in \mathcal{N}$, $i \in \mathcal{I}$ and $t \in [T]$. For ease of notation, let us define $B := \sqrt{8 \log(NMK/\delta)}$ and $w_{uik}^t = \sqrt{1/n_{uik}^t}$. Assuming that this condition holds, we can show that

$$\rho(\boldsymbol{\pi}_K^*) - \rho(\boldsymbol{\pi}) \leq \sum_{t=1}^{T} \left[ \max_{\boldsymbol{X} \in \mathcal{X}_e^t} \sum_{u \in \mathcal{N}} \sum_{i \in \mathcal{I}} \sum_{k=1}^{K} x_{uik} \psi_{uik} - \sum_{u \in \mathcal{N}} \sum_{i \in \mathcal{I}} \sum_{k=1}^{K} x_{uik}^t \psi_{uik} \right]$$

$$\leq \sum_{t=1}^{T} \left[ \max_{\boldsymbol{X} \in \mathcal{X}_e^t} \sum_{u \in \mathcal{N}} \sum_{i \in \mathcal{I}} \sum_{k=1}^{K} x_{uik} \left( \widehat{\psi}_{uik}^t + \frac{B}{w_{uik}^t} \right) - \sum_{u \in \mathcal{N}} \sum_{i \in \mathcal{I}} \sum_{k=1}^{K} x_{uik}^t \psi_{uik} \right]$$

$$\leq \sum_{t=1}^{T} \left[ \sum_{u \in \mathcal{N}} \sum_{i \in \mathcal{I}} \sum_{k=1}^{K} x_{uik}^t \left( \widehat{\psi}_{uik}^t + \frac{B}{w_{uik}^t} \right) - \sum_{u \in \mathcal{N}} \sum_{i \in \mathcal{I}} \sum_{k=1}^{K} x_{uik}^t \psi_{uik} \right]$$

$$\leq \sum_{t=1}^{T} \sum_{u \in \mathcal{N}} \sum_{i \in \mathcal{I}} \sum_{k=1}^{K} x_{uik}^t \left( \widehat{\psi}_{uik}^t - \psi_{uik} + \frac{B}{w_{uik}^t} \right).$$

Hence, with probability $1 - \delta$, we have

$$\rho(\boldsymbol{\pi}_K^*) - \rho(\boldsymbol{\pi}) \leq 2B \sum_{t=1}^{T} \sum_{u \in \mathcal{N}} \sum_{i \in \mathcal{I}} \sum_{k=1}^{K} \frac{x_{uik}^t}{w_{uik}^t}. \tag{23}$$

At each time step $t$, consider the list consisting of $w_{uik}^t$ for all $(u, i, k) \in \mathcal{O}_t := \{(u, i, k) : x_{uik}^t = 1\}$. Let's now consider the overall list consisting of the concatenation of all of these lists over all rounds. Let's order this list in decreasing order to obtain a list $\widetilde{w} = (\widetilde{w}_1, \widetilde{w}_2, \ldots, \widetilde{w}_J)$ where $J = \sum_{t=1}^{T} |\mathcal{O}_t| \leq LT$. Using this notation, we have

$$\rho(\boldsymbol{\pi}_K^*) - \rho(\boldsymbol{\pi}) \leq 2B \sum_{j=1}^{J} \widetilde{w}_j.$$

First, for any $(u, i, k)$ pair, the list $\widetilde{w}$ can contain at most $1 + 1/c^2$ elements that are associated with $(u, i, k)$ and larger than or equal to $c$. Secondly, note that $x_{uik}^t = 1$ only if $B/w_{uik}^t \geq \max_k \psi_{uik} - \psi_{uik} = \Delta_{uik}$. As a result, size of $\{t : x_{uik}^t = 1\}$ is at most $B^2/\Delta_{uik}^2$. Therefore, the number of times a $(u, i, k)$ pair can appear in the list $\widetilde{w}$ is also upper bounded by $B^2/\Delta_{uik}^2$. Therefore, summing over all $k \in [K]$, we can upper bound the number of elements that are associated

with $(u, i)$ and larger than or equal to $c$ by

$$\sum_{k=1}^{K} \min\left\{1 + \frac{1}{c^2}, \frac{B^2}{\Delta_{uik}^2}\right\} = \sum_{k=0}^{K-1} \min\left\{1 + \frac{1}{c^2}, \frac{B^2}{\widetilde{\Delta}_{uik}^2}\right\}$$

$$\leq 1 + \frac{1}{c^2} + \sum_{k=1}^{K-1} \min\left\{1 + \frac{1}{c^2}, \frac{B^2}{C_1^2}\left(\frac{2K}{k}\right)^4\right\}$$

$$\leq 1 + \frac{1}{c^2} + \frac{B}{C_1}\sqrt{1 + \frac{1}{c^2}}\sum_{k=1}^{K-1}\left(\frac{2K}{k}\right)^2$$

$$= O\left(1 + \frac{1}{c^2} + BK^2\sqrt{1 + \frac{1}{c^2}}\right).$$

where the second step uses Lemma 11. Thus, the total number of times that any confidence set can have size at least $\widetilde{w}_j$ is upper bounded by $O\left(NM\left(1 + \frac{1}{\widetilde{w}_j^2}\right) + NMBK^2\sqrt{1 + \frac{1}{\widetilde{w}_j^2}}\right)$. Using this result, we can write $\widetilde{w}_j = O\left(\min\left\{1, \frac{1}{\sqrt{(j/BNMK^2)^2 - 1}} + \frac{1}{\sqrt{j/NM - 1}}\right\}\right)$. Hence,

$$\sum_{j=1}^{J} \widetilde{w}_j = O\left(\sum_{j=1}^{J} \min\left\{1, \frac{1}{\sqrt{(j/BNMK^2)^2 - 1}} + \frac{1}{\sqrt{j/NM - 1}}\right\}\right)$$

$$= O\left(\sum_{j=1}^{LT} \min\left\{1, \frac{1}{\sqrt{(j/BNMK^2)^2 - 1}} + \frac{1}{\sqrt{j/NM - 1}}\right\}\right)$$

$$= O\left(BNMK^2 \log(LT) + \sqrt{NMLT}\right).$$

Therefore,

$$\rho(\boldsymbol{\pi}_K^*) - \rho(\boldsymbol{\pi}) = O\left(B^2 NMK^2 \log(LT) + \sqrt{NMLT}\right). \tag{24}$$

Next, we bound $\rho(\boldsymbol{\pi}^*) - \rho(\boldsymbol{\pi}_K^*)$. Note that for each $(u, i)$, at least one of the numbers $\{1/K, 2/K, \ldots, 1\}$ lies within $1/K$ of $p^*(u, i)$. Then, by Lemma 12, $\psi_{ui}^* - \max_k \psi_{uik} \leq C_2/K^2$ for some absolute constant $C_2$. Therefore, the gap $\rho(\boldsymbol{\pi}^*) - \rho(\boldsymbol{\pi}_K^*)$ is upper bounded as

$$\rho(\boldsymbol{\pi}^*) - \rho(\boldsymbol{\pi}_K^*) \leq \frac{C_2 LT}{K^2}. \tag{25}$$

Combining equations (24) and (25), we have

$$\mathcal{R}(T, \boldsymbol{\pi}) = O\left(B^2 NMK^2 \log(LT) + \sqrt{NMLT} + \frac{LT}{K^2}\right).$$

Lastly, we choose $K^4 = \Theta\left(\frac{LT}{NM \log(LT)}\right)$ to obtain

$$\mathcal{R}(T, \boldsymbol{\pi}) = O\left(B^2 \sqrt{NMLT \log(LT)}\right)$$

$$= O\left(\sqrt{NMLT \log(LT) \log^2(NMK/\delta)}\right)$$

$$= O\left(\sqrt{NMLT \log(LT) \log^2(NMT/\delta)}\right).$$

where the last step uses $0 \leq K^4 \leq T$ and $T \geq 1$.

### D.3 Proof of Theorem 6

Consider the same market setting (demands and endowments) in Appendix Section B.2, but assume that the valuations are i.i.d. random instead of being fixed. By construction, the optimum offering pattern at each time is to offer all available items to a single user that has non-zero demand. Hence, the problem of the provider reduces to only learning the price at which it should offer each item. There are $NM$ actual user-item pairs and each item $i \in [M]$ is offered to user $u \in [N]$ for $\Theta(LT/NM')$ rounds. In the literature on pricing optimization with i.i.d. valuations, each pricing problem is known to have $\Omega(\sqrt{T_o})$ regret in $T_o$ rounds (Kleinberg and Leighton, 2003). Therefore, any policy must have $\Omega(\sqrt{NMLT})$ regret in total.

## E   Martingale Exponential Inequalities

Consider a sequence of random variables $(Z_n)_{n \in \mathbb{N}}$ adapted to the filtration $(\mathcal{H}_n)_{n \in \mathbb{N}}$. Assume $\mathbb{E}[\exp(\lambda Z_i)]$ is finite for all $\lambda$. Define the conditional mean $\mu_i = \mathbb{E}[Z_i | \mathcal{H}_{i-1}]$, and define the conditional cumulant generating function of the centered random variable $[Z_i - \mu_i]$ by $\psi_i(\lambda) := \log \mathbb{E}[\exp(\lambda[Z_i - \mu_i]) | \mathcal{H}_{i-1}]$. Then, for a process $(M_n(\lambda))_{n \in \mathbb{N}}$ defined as

$$M_n(\lambda) = \exp \left\{ \sum_{i=1}^{n} \lambda[Z_i - \mu_i] - \psi_i(\lambda) \right\},$$

we can prove the following properties.

**Lemma 13.** $(M_n(\lambda))_{n \in \mathbb{N}}$ *is a martingale with respect to the filtration* $(\mathcal{H}_n)_{n \in \mathbb{N}}$*, and* $\mathbb{E}[M_n(\lambda)] = 1$ *for any* $\lambda$.

*Proof.* By definition, we have

$$\mathbb{E}[M_1(\lambda) | \mathcal{H}_0] = \mathbb{E}[\exp\{\lambda[Z_1 - \mu_1] - \psi_1(\lambda)\} | \mathcal{H}_0] = 1.$$

Then, for any $n \geq 2$,

$$\begin{aligned}
\mathbb{E}[M_n(\lambda) | \mathcal{H}_{n-1}] &= \mathbb{E}[M_{n-1}(\lambda) \exp\{\lambda[Z_n - \mu_n] - \psi_n(\lambda)\} | \mathcal{H}_{n-1}] \\
&= M_{n-1}(\lambda) \mathbb{E}[\exp\{\lambda[Z_n - \mu_n] - \psi_n(\lambda)\} | \mathcal{H}_{n-1}] \\
&= M_{n-1}(\lambda),
\end{aligned}$$

since $M_{n-1}(\lambda)$ is a measurable function of the filtration $\mathcal{H}_{n-1}$. $\qquad \square$

**Lemma 14.** *For all* $x \geq 0$ *and* $\lambda \geq 0$,

$$\mathbb{P} \left( \sum_{i=1}^{t} \lambda Z_i \leq x + \sum_{i=1}^{t} [\lambda \mu_i + \psi_i(\lambda)] \quad , \forall t \in \mathbb{N} \right) \geq 1 - e^{-x}.$$

*Proof.* For any $\lambda$, $(M_n(\lambda))_{n \in \mathbb{N}}$ is a martingale with respect to $(\mathcal{H}_n)_{n \in \mathbb{N}}$ and $\mathbb{E}[M_n(\lambda)] = 1$ by Lemma 13. For arbitrary $x \geq 0$, define $\tau_x = \inf\{n \geq 0 | M_n(\lambda) \geq x\}$ and note that $\tau_x$ is a stopping time corresponding to the first time $M_n$ crosses the boundary $x$. Since $\tau$ is a stopping time with respect to $(\mathcal{H}_n)_{n \in \mathbb{N}}$, we have $\mathbb{E}[M_{\tau_x \wedge n}(\lambda)] = 1$. Then, by Markov's inequality

$$x\mathbb{P}(M_{\tau_x \wedge n}(\lambda) \geq x) \leq \mathbb{E}[M_{\tau_x \wedge n}(\lambda)] = 1.$$

Noting that the event $\{M_{\tau_x \wedge n}(\lambda) \geq x\} = \bigcup_{k=1}^{n} \{M_k(\lambda) \geq x\}$, we have

$$\mathbb{P} \left( \bigcup_{k=1}^{n} \{M_k(\lambda) \geq x\} \right) \leq \frac{1}{x}.$$

Taking the limit as $n \to \infty$, and applying monotone convergence theorem shows that

$$\mathbb{P} \left( \bigcup_{k=1}^{\infty} \{M_k(\lambda) \geq x\} \right) \leq \frac{1}{x}.$$

Consequently, we can write

$$\mathbb{P}\left(\bigcup_{k=1}^{\infty}\{M_k(\lambda) \geq e^x\}\right) \leq e^{-x}.$$

Then, by definition of $M_k(\lambda)$, we conclude

$$\mathbb{P}\left(\bigcup_{k=1}^{\infty}\left\{\sum_{i=1}^{n}\lambda[Z_i - \mu_i] - \psi_i(\lambda) \geq x\right\}\right) \leq e^{-x}$$

$\square$

