# OpenReview forum: "Online Pricing for Multi-User Multi-Item Markets"
_NeurIPS.cc/2023/Conference — NeurIPS 2023 poster_

### Official Review · Reviewer_bZxp · 2023-06-22

**Soundness:** 2 fair
**Presentation:** 2 fair
**Contribution:** 2 fair
**Rating:** 5
**Confidence:** 3

**Summary:**

The paper proposes online pricing and allocation strategies for multi-user multi-markets model under three different valuation models.

The main contribution is to extend dynamic pricing strategies from the literature to the case of more than one item and more than one user.

The setting goes as follows:
- At every iteration, each user $u$ presents a maximum demand for items $d_u^t$, that is observed.
- the algorithm allocates items and prices.
- a user accepts the items for which his valuation is higher than the offered price, accepting at most $d_u^t$ items.

The study the fixed valuation model in which the valuation of users for each item is fixed and deterministic, the random experience model, where a user's valuation for an item is the mean of his past experiences, and the random valuation model where the valuations are drawn from fixed but unknown distributions.

Presented algorithms construct confidence intervals for the user-item valuations. At every iteration, the items are allocated according to the so-called optimism principle, meaning the optimal allocation for the upper bounds on the user-item valuations is chosen.

The price set for the item depends on the valuation model.

**Strengths:**

The paper is clear and the algorithms well presented for the most part. The study of larger markets than one item/one user certainly is interesting and relevant for applications.

**Weaknesses:**

My main criticism concerns the way the paper is written. A lot of emphasis is put on the pricing strategies. However, at least two of the method for pricing, for the fixed valuation model and the random valuation model, already exist in the literature (reference 3) and this is not discussed in the paper.
More precisely, the methods of geometrically decreasing intervals and discretization of the price interval are exactly the methods used in ref 3.

There are some unclarities in the proofs, as listed below.

1. The proof of Theorem 1 is not clear at all. Parameter $\epsilon$ is set to $1/LT$ without explanation. This can only be done because exploration stops when the interval for the valuation becomes small enough. But although it appears in the pseudo-code of the algorithm, it is not evoked in the main text nor in the proof. However, this feature of stopping exploration after some time is crucial in avoiding linear regret. This is clear for a reader familiar with ref 3, but I do not believe it would be otherwise.

2. In Appendix C, $y_ui$ is used without introduction. In figure 2, notations are also used before they are introduced.

3. Line 494, the lists are defined too informally.




**Questions:**

Overall, the paper has interesting elements, but a good discussion on the dependence of the problem in the parameter L, M and N is missing, as this is the main contribution of the paper.

Precise references to where previously existing methods were used are missing. Adding this would help the reader understand the contribution of the paper better.

On a side note, the assumption that $d_u^t$ is observed seems rather strong, if the authors have a good argument as to why it is necessary (or not), it would be interesting to add.

---

> ### Author Rebuttal · Authors · 2023-08-09
>
> Thank you for your constructive feedback and thoughtful questions.
>
> In this paper, our main contribution is the analysis of revenue-maximizing algorithms that allocate and price multiple items to multiple users. As stated in our related work section, Kleinberg and Leighton [3] developed a widely-appreciated framework to maximize revenue from selling copies of a single good to sequentially arriving users. Their scenario corresponds to a special case of our problem where the provider interacts with only one user at each round and sells a single type of item. In contrast to this setting, a provider that simultaneously interacts with multiple users faces additional challenges in offering and pricing the items; such challenges are addressed in our work.
>
> The hardest challenge is to decide on simultaneous allocation and pricing actions that can balance exploration with exploitation. At a high level, we tackle this challenge by combining ideas from the dynamic pricing and combinatorial bandits literature. The offers are chosen using the OFU principle as it is often done in combinatorial bandits literature. On the other hand, our solutions for pricing the items are based on the foundational techniques presented in [3]. Regarding related comments, we will provide more precise references to [3] in Sections 4.1 and 4.3 with the goal of improving clarity.
>
> Please find our responses to your other questions below:
> * In Section 1.1 and in lines 166-168, we already have some discussion relating to the problem's dependency on the parameters $N$, $M$, and $L$. However, in response to related comments, we want to expand on that discussion by including the following paragraphs. \
> Under all three valuation models, we observe that optimal regret rates depend on two central quantities: $NM$ and $LT$. The $NM$ term in these expressions corresponds to the number of user-item pairs for which we need to learn the valuations. On the other hand, the $LT$ factor in the optimal regret rate corresponds to the maximum total number of offers that we make. \
> For the specific case of the random valuations model, it is possible to draw further conclusions about the complexity of the problem. Recall that we achieve an almost optimal regret rate of $\widetilde{O}(\sqrt{NMLT})$. In comparison, consider a social-welfare maximization problem where we can make real-valued observations of the random valuations $v_{ui}^t$ for each allocated user-item pair and our goal is to maximize the sum of expected valuations for our allocations. As this problem can be easily modeled as a semi-bandit problem with $NM$ arms of which at most $L$ can be chosen simultaneously at each of the $T$ rounds, the optimum regret rate would be $O(\sqrt{NMLT})$ (Kveton et al., 2015). Hence, we can conclude that our revenue maximization problem with accept/reject feedback is no harder than the social welfare maximization problem with real-valued random reward feedback.
> * Our focus is on the problem of allocation and pricing to maximize long-term revenue when user preferences are unknown. Applications relevant to our study include online delivery, hotel bookings, ride-shares, etc., where users come to the platform in search of products or services. In these examples, we can readily consider that users demand only one item ($d^t_u = 1$) when they are active and they have no demand ($d^t_u = 0$) when inactive. While this assumption might be too strong for some applications, our work serves as a valuable starting point for investigation.
> Please find our responses regarding other unclarities below:
> * We will include the following paragraph after line 227 to elaborate on the need to stop exploration.\
> Due to this update rule for $\beta_{ui}$, the length of the corresponding search interval $b_{ui} - a_{ui}$ is at most $\sqrt{\beta_{ui}}$ at any given time. Let an epoch for a user-item pair $(u, i)$ refer to the set of all time steps in which user $u$ is offered item $i$ using the same price increment $\beta_{ui}$. Since the algorithm operates by offering item $i$ to user $u$ at prices increasing by $\beta_{ui}$, the number of queries in an epoch can be anywhere between 1 and $1/\sqrt{\beta_{ui}}$. In the proof of Theorem 1, we show that the regret we incur over a single epoch for a user-item pair $(u, i)$ is $O(1)$, and the number of exploratory epochs can be as large as $T$ in the worst case. Therefore, continuing to explore indefinitely would result in linear regret. To avoid this issue, the algorithm should stop exploration after achieving a certain level of precision and offer item $i$ to user $u$ at the maximum price that is certainly acceptable, namely $a_{ui}$.
>
> * To address the weakness (2), we will use $v_{ui}^t$ instead of $y_{ui}$ throughout the proof.
>
> * Regarding weakness (3), we will revise the rest of the proof after line 494 as follows. \
> \begin{align}
> \sum_{u\in\mathcal{N}}\sum_{i \in \mathcal{I}}\sum_{t:x_{ui}^t = 1}\sqrt{\frac{1}{m_{ui}^t}}\leq \sum_{u\in\mathcal{N}}\sum_{i\in\mathcal{I}}\left(\sum_{t:x_{ui}^t=1\text{ and }\nexists j\in\mathbb{N}:n_{ui}^t=2^j}\sqrt{\frac{1}{m_{ui}^t}}+\sum_{t:x_{ui}^t=1\text{ and }\exists j\in\mathbb{N}:n_{ui}^t=2^j}\sqrt{\frac{1}{m_{ui}^t}}\right)\leq\sum_{u\in\mathcal{N}}\sum_{i\in\mathcal{I}}\left(2\sqrt{m_{ui}^T}+\log T\right)\end{align}
> where we upper bound the first term using $\sum_{k=1}^{n}\frac{1}{\sqrt{k}}\leq\int_{0}^{n}\frac{\mathrm{d}x}{\sqrt{x}}=2\sqrt{n}$ and the second term by noting that it can have at most $\log T$ terms. Also, note that the total number of offers over all time intervals is upper bounded by $\sum_{u\in\mathcal{N}}\sum_{i\in\mathcal{I}}m_{ui}^T\leq LT$. Then, using the Cauchy-Schwartz inequality,
> \begin{align}
> \sum_{u\in\mathcal{N}}\sum_{i \in \mathcal{I}}\sum_{t:x_{ui}^t = 1}\sqrt{\frac{1}{m_{ui}^t}}\leq 2\sqrt{NM\sum_{u\in\mathcal{N}}\sum_{i\in\mathcal{I}}m_{ui}^T}+NM\log T\leq O(\sqrt{NMLT}+NM\log T).
> \end{align}
> Putting all together, we conclude the proof of the lemma.

---

> > ### Comment · Reviewer_bZxp · 2023-08-15
> >
> > Thank you for those clarifications. I have decided to slightly improve my score, as my main criticism concerned the presentation of the paper and the inclusion of the contributions within the literature, and those paragraphs resolve that concern.

---

### Official Review · Reviewer_2rbX · 2023-07-01

**Soundness:** 3 good
**Presentation:** 4 excellent
**Contribution:** 3 good
**Rating:** 7
**Confidence:** 3

**Summary:**

The paper considers the online pricing problem where there are several users and items. Each user has a (private) valuation for each item and a (public) upper bound for the number of items desired, both of which could vary at different rounds. The provider has an available item set at each round, and the goal is to distribute them to the users such that the total revenue is maximized. The paper considers three settings for the uses' valuation functions: (1) fixed valuations, (2) random experiences, and (3) random valuations. For each setting, the corresponding algorithm is proposed and analyzed, along the theoretical lower bound. For the first two settings, the basic algorithmic idea is trying to learn an accurate valuation interval for each item-user pair, and then computing the assignment and the prices according to the estimated intervals; while for the last setting, the authors quantize the set of possible prices and then borrow the idea from the multi-armed bandit problem to give an algorithm. Finally, the proposed algorithms are evaluated empirically.

**Strengths:**

- The paper makes theoretical contributions. The multi-user multi-item online pricing problem introduced makes sense. The authors consider different valuation settings of this problem, and obtain small gaps between the upper bounds and the lower bounds. The analysis is non-trival.

- The paper is well-written. For each algorithm, the authors always provide the basic algorithmic intuition, which can help the reader understand the algorithm.

**Weaknesses:**

- The discussion of the algorithms' running time is missing. Further, in the current statement, the algorithms need to solve linear programs at each round, but essentially, this can be formulated into a bipartite matching instance and thus, the classical matching algorithm can be applied and help to improve the running time.

**Questions:**

Minor comments:

- It might be better to point out that $\lor$ and $\land$ represent "taking max" and "taking min" respectively in the paper.

- In the statement of algorithm 3, $n_{uik}$ is initialized by 0, but in the first iteration, we compute the term $\sqrt{8\log (NMKT) / n_{uik}}$.

**Limitations:**

I didn't see any negative societal impact of the paper.

---

> ### Author Rebuttal · Authors · 2023-08-09
>
> Thank you for your constructive comments and questions. Please find our responses to the questions below:
>
> * Based on your comments, we agree to include a discussion on the computational complexity and resource requirements of the algorithms. As you noted, the integer linear program in (7) can be written as an instance of maximum weight bipartite matching. Then, using a variant of the Hungarian algorithm for unbalanced bipartite graphs (Ramshaw and Tarjan, 2012), this problem can be solved in space $O(NM)$ and time $O(NML)$ in the worst case. As this result shows, for a fixed number of items, space and time complexities both have a linear dependency on the number of users.\
> **Details of the construction:** In this graph, we represent each user $u$ with $d_u^t$ left nodes and we represent each endowed item $i$ with a right node. Then, we construct the complete weighted bipartite graph where the weight of an edge between a node for user $u$ and a node for item $i$ is given as $v_{ui}$. In total, this graph has $D_t$ left vertices, $E_t$ right vertices, and $D_t E_t$ weighted edges where $D_t$ and $E_t$ correspond to total demand and endowment at time $t$ (as given in Definition 2). Since we can upper bound $D_t \leq N$, $E_t \leq M$, and $\min \\{D_t, E_t\\} \leq L$, the results in the previous paragraph follow.\
> **Reference:** Ramshaw, L., & Tarjan, R. E. (2012). On Minimum-Cost Assignments in Unbalanced Bipartite Graphs.
> * We agree that it will be better to clearly define notations for “taking min” and “taking max” in our notation section at line 120.
> * The division by zero in Algorithm 3 can be interpreted as resulting in infinity. Since we set $b_{uik}$ as the minimum of $( \psi_{uik} + \sqrt{8 \log (NMKT)/n_{uik}} )$ and $1$, we always set $b_{uik} = 1$ when $n_{uik} = 0$. For clarity, we will include a footnote that briefly explains how to interpret $\sqrt{8 \log (NMKT)/n_{uik}}$ when $n_{uik} = 0$.

---

> > ### Comment · Reviewer_2rbX · 2023-08-16
> >
> > I have gone through the rebuttal. The reduction stated makes sense to me.

---

### Official Review · Reviewer_PhX7 · 2023-07-09

**Soundness:** 3 good
**Presentation:** 3 good
**Contribution:** 3 good
**Rating:** 5
**Confidence:** 4

**Summary:**

The paper studies the problem of dynamic pricing in which multiple items are offered to multiple users at each round. Authors propose a novel algorithm for maximizing revenue under three user valuation models: fixed valuations,  random experiences and random valuations. Authors provide theoretical guarantees and provide simulation results supporting the theoretical claims.

**Strengths:**

**Strengths**

1.) The paper is well written and easy to follow. Authors motivate the problem formulation well.

2.) Authors provide a detailed related work section and contributions well positioned in the relevant literature.

3.) The theoretical results appear to be sound and the paper also provide novel algorithmic contributions.


**Weaknesses:**

**Weaknesses**

1.) The main weakness of the paper is lack of comparison with existing methods. Authors do not provide a discussion comparing their theoretical or simulation results with existing work.

2.) Including a proof sketch in the main paper will improve the quality of the paper.

3.) Including additional simulation results will improve the paper.


**Questions:**

1.) Authors assume that items that are not sold in one round can not be stored to be sold in future rounds. What are technical challenges associated with relaxing this assumption?

2.) Can this results be extended to the case where users only accept an item if its price is lower than the valuation by some

**Limitations:**

The paper does not discuss the limitations of the method and potential societal impact.

---

> ### Author Rebuttal · Authors · 2023-08-09
>
> Thank you for your constructive comments and questions. Please find our responses to the questions below:
>
> * As discussed in our related works section, dynamic pricing literature has only considered settings where a single user interacts with the provider per time step. Therefore, even if their valuation models are similar, those theoretical or empirical results are not directly comparable with ours. To best position our work within the literature, in our related works section, we include bodies of works that are most relevant to the framework that we are proposing. Throughout that section, we explain the similarities and differences between our work and cited papers. \
> Though it’s not a direct comparison, it is possible to compare our work with Kleinberg and Leighton [3] since our results can be interpreted as a generalization of their theoretical findings. This is because their market model is a special case of ours where the number of users is $N=1$, the number of items is $M=1$, and the load is $L=1$. For fixed valuations, we achieve almost optimal regret of order $O(NM\log\log(LT))$ which reduces to $O(\log \log T)$ in the single-user single-item case. Similarly, for the random valuations model, we achieve regret of order $O(\sqrt{NMLT})$ which reduces to $O(\sqrt{T})$ in the single-user single-item case. For both cases, these results match the results of Kleinberg and Leighton [3].
>
> * We provide discussions on the reasoning behind our allocation and pricing mechanisms throughout Sections 4.1, 4.2, and 4.3 within the limitations of space constraints. Based on your comment, we will extend these discussions to speak about the high-level details of the proof as well.
>
> * In our current framework, the demand and endowment information for time $t$ is only made available to the provider at time $t$. However, if the provider can store the items to sell in future rounds, it also needs to have some idea about the demand for future rounds to be able to determine how much it needs to store. To formulate a no-regret algorithm under this scenario, one would need to analyze the tradeoff between offering available items to learn user valuations faster and storing the items to sell in future rounds at possibly higher prices. We believe that the combination of learning user preferences and estimating future demand/endowment is an important direction and we believe that the current work serves as a foundation for exploring this topic.
>
> * If we were to consider a case where users only accept an item if its price is lower than the valuation by some fixed (but unknown) amount $c_{ui}$, then our results would be still valid for all valuation models. Letting $b_{ui}^t = v_{ui}^t - c_{ui}$ to be the acceptance/rejection threshold and replacing $v_{ui}^t$ with $b_{ui}^t$ in the definitions of optimality and regret, all three algorithms would achieve the same order of regret without any modification.

---

### Official Review · Reviewer_Hkdu · 2023-07-24

**Soundness:** 3 good
**Presentation:** 3 good
**Contribution:** 3 good
**Rating:** 6
**Confidence:** 4

**Summary:**

The paper proposes algorithms for optimizing the sale of multiple goods to multiple users, taking into account their time-varying valuations throughout repeated rounds. These algorithms efficiently learn from users' accept or reject feedback and utilize this information to make optimal offers and prices based on three user valuations: fixed valuation, random experience, and random valuation. To evaluate the effectiveness of the proposed algorithms, the paper provides revenue regret guarantees. Notably, the significant contribution of this paper lies in the introduction of the concept of user's random experience as a crucial valuation to consider in dynamic pricing algorithms.

**Strengths:**

The paper addresses the problem of revenue maximization in a dynamic market with time-varying valuations. Specifically, the paper introduces the concept of "random experience", which is a novel and unexplored idea in prior literature.
The paper offers a rigorous theoretical analysis of the proposed algorithms and assesses their effectiveness and efficiency in practice by evaluating nearly-optimal revenue regret guarantees for each algorithm.
The paper exhibits a well-organized structure, presenting the problem formulation, proposed algorithms, theoretical analysis, and experimental results in a clear and coherent manner.


**Weaknesses:**

Some statements are general and lack support or explicit example, such as in line 17, which states, “The ability to design algorithms that can achieve the optimal sale of goods to multiple users having time-varying valuations for each of the goods is both timely and relevant, given the explosion in the use of data in large-scale systems.” How does the author come to this statement? I would suggest briefly citing more literature to support the argument.
Figure 1 is unclear, and it is challenging to understand how the provider learns user valuation from the current depiction. Additionally, the meaning of the gray user representative is not apparent. To improve the figure's clarity, I recommend providing a more detailed description.
The paper may lack comparisons with relevant literature. The author mentions that fixed valuation and random valuation have been used as standard models for dynamic pricing in previous research. Therefore, I would suggest that the author could further elaborate on and compare this paper’s results with previous work. In comparison with previous works, the author can emphasize the advantages and differences of their proposed method, such as differences in experimental design or interpretation of results. This will help clarify the innovation and contribution of the paper.


**Questions:**

In line 56, why are the algorithms focused on offering each item to only one user during each round? The framework of "multi-user multi-item" proposed in the paper suggests that interactions happen between multiple users and multiple items over repeated rounds, indicating a dynamic market setting. While the constraint of offering one item to one user during each round simplifies the problem, it may not fully reflect the dynamic market. Therefore, I would like to see that the algorithms can be focused on more items to one user, one item to more users, or more items to more users, which aligns better with the multi-user multi-item market framework proposed in the paper.
In Section 5, why are two different datasets used for numerical experiments? Figure 3 is based on 20 sets of experiments with N=150 users and M=100 items, while Figure 4 is based on 5 sets of experiments with N=30 users and M=20 items.

---

> ### Author Rebuttal · Authors · 2023-08-09
>
> Thank you for your constructive comments and questions. Please find our responses to the questions below:
>
> * As discussed in our related works section, dynamic pricing literature has only considered settings where a single user interacts with the provider per time step. Therefore, even if their valuation models are similar, those algorithms and theoretical results are not directly comparable with ours. To best position our work within the literature, the related works section includes bodies of works that are most relevant to the framework that we are proposing. Throughout that section, we explain the similarities and differences between our work and cited papers. \
> Though it’s not a direct comparison, it is possible to compare our work with Kleinberg and Leighton [3] since our results can be interpreted as a generalization of their theoretical findings. This is because their market model is a special case of ours where the number of users is $N=1$, the number of items is $M=1$, and the load is $L=1$. For fixed valuations, we achieve almost optimal regret of order $O(NM\log\log(LT))$ which reduces to $O(\log \log T)$ in the single-user single-item case. Similarly, for the random valuations model, we achieve regret of order $O(\sqrt{NMLT})$ which reduces to $O(\sqrt{T})$ in the single-user single-item case. For both cases, these results match the results of Kleinberg and Leighton [3].
>
> * We do have multiple active users in each round and we allocate multiple items across them simultaneously. Our assumption is only about not assigning the same item to multiple users in any round. Nevertheless, our framework can also be readily extended to capture settings in which multiple users are offered the same item. We could achieve this by replicating each item according to its number of available copies. Note that this would correspond to an increase in the number of items as we will need to count each copy as a separate item. For all three valuation models, the corresponding algorithms would achieve regret upper bounds that hold with this modification to the number of items. To not complicate the market model and its analysis in this work, we only consider the case we are given a single copy of each item.
>
> * We chose relatively large numbers of users and items for experiments in Figure 3 to show that our algorithms can achieve diminishing regret over time for larger-scale problems. On the other hand, for the experiments in Figure 4, we opted for a smaller problem because we needed to run many more experiments corresponding to each different time horizon. For instance, we run 20 experiments with $T=2^{10}$ for the first plot in Figure 3 while we run 50 experiments with up to $T=2^{14}$ for the first plot in Figure 4. \
> Nevertheless, we have improved the time complexity of our simulations using a maximum weight matching algorithm instead of an LP solver as suggested by reviewers 2rbX and ABaL. Therefore, we are now able to run the experiments in a shorter time and generate a figure showing “regret as a function of time horizon T under different valuation models with N = 150 users and M = 100 items”. (Please see the pdf document in our global response for this figure.) In our revised manuscript, we are committed to repeating this experiment across multiple runs and replacing Figure 4 with those results.
>
> * We appreciate your feedback regarding the motivating statements and their need for explicit support. We are fully committed to addressing this concern by incorporating relevant citations that strengthen the rationale behind these statements. This effort will significantly enhance the credibility and context of our arguments.
>
> * In relation to Figure 1, we recognize your observation regarding its clarity. We deeply value your input and are committed to incorporating your feedback as we revise the figure. This will involve providing a more detailed description that better elucidates the process and ensures a clearer understanding for our readers.

---

### Official Review · Reviewer_ABaL · 2023-07-24

**Soundness:** 3 good
**Presentation:** 2 fair
**Contribution:** 3 good
**Rating:** 5
**Confidence:** 3

**Summary:**

This paper addresses the problem of online pricing in multi-user multi-item markets. The main objective is to maximize revenue by selling multiple items to multiple users in each round. The paper proposes algorithms that efficiently offer and price items while learning user valuations from accept/reject feedback. It considers three user valuation models and provides algorithms with nearly-optimal revenue regret guarantees. Additionally, the paper introduces a problem-dependent load parameter and designs regret-optimal algorithms for different market settings. Overall, the paper makes contributions in terms of user valuation models, load parameter analysis, and algorithm design, being the first to address this problem.

**Strengths:**

This paper introduces a novel problem of online pricing in multi-user multi-item markets and addresses the specific challenge of maximizing revenue by selling multiple items to multiple users in each round. The quality of the paper is commendable, with a comprehensive analysis of different user valuation models, well-explained methodology, and rigorously analyzed algorithms. The paper is also clearly written and well-structured, making it easy for readers to follow the flow of ideas. Overall, the paper demonstrates originality in problem formulation, high-quality research methodology, clear presentation of ideas, and significant contributions to the field.

**Weaknesses:**

1.	The paper makes a simplifying assumption that each item is offered to only one user during each round. However, this assumption may not capture the realistic settings in which multiple users may have overlapping preferences for the same item and providers may have incentives to offer the same item to multiple users to increase their revenue.
2.	The paper does not investigate the scalability of the proposed algorithms. It would be important to evaluate how the algorithms scale with respect to the number of users, items, and rounds. This evaluation would reveal the computational complexity and resource requirements of the algorithms in large-scale multi-user multi-item markets.
3.	The paper does not discuss practical considerations that may arise in real-world implementations, such as privacy concerns, fairness considerations, or strategic user behavior. It would be beneficial to address these practical considerations and discuss how the proposed algorithms can cope with such challenges or potential extensions to handle them.


**Questions:**

Please refer to weaknesses 1, 2, and 3

**Limitations:**

The paper presents several limitations that could be addressed in future work. Firstly, the assumption that each item is offered to only one user during each round may not reflect realistic settings where multiple users may have overlapping preferences for the same item. Secondly, the paper does not address practical considerations such as privacy concerns, fairness considerations, or strategic user behavior. Lastly, the exploration of different market settings is limited, and it would be interesting to analyze the performance of the proposed algorithms in diverse market scenarios. Addressing these limitations would provide a more comprehensive understanding of the applicability and robustness of the algorithms.

---

> ### Author Rebuttal · Authors · 2023-08-09
>
> Thank you for your constructive comments and questions. Please find our responses to the questions below:
>
> * Our framework can be readily extended to capture settings in which multiple users are offered the same item. We could achieve this by replicating each item according to its number of available copies. Note that this would correspond to an increase in the number of items as we will need to count each copy as a separate item. For all three valuation models, the corresponding algorithms would achieve regret upper bounds that hold with this modification to the number of items. To not complicate the market model and its analysis in this work, we only consider the case we are given a single copy of each item. \
> On the other hand, another possibility is to offer an item to more users than the number of copies we have. However, this also brings with it the risk of demand exceeding capacity. If we want to allow algorithms that take such risks, we would need to (1) estimate the chances of overselling and (2) make quantifying assumptions about the consequences of not being able to satisfy users' requests. We agree that relaxing this assumption is an intriguing and important area for future work, and we believe that the current work serves as a foundation for exploring this topic.
>
> * Based on your comments, we agree to include a discussion on the computational complexity and resource requirements of the algorithms. The integer linear program in (7) can be written as an instance of maximum weight bipartite matching. Then, using a variant of the Hungarian algorithm for unbalanced bipartite graphs (Ramshaw and Tarjan, 2012), this problem can be solved in space $O(NM)$ and time $O(NML)$ in the worst case. As this result shows, for a fixed number of items, space and time complexities both have a linear dependency on the number of users. \
> **Details of the construction:** In this graph, we represent each user $u$ with $d_u^t$ left nodes and we represent each endowed item $i$ with a right node. Then, we construct the complete weighted bipartite graph where the weight of an edge between a node for user $u$ and a node for item $i$ is given as $v_{ui}$. In total, this graph has $D_t$ left vertices, $E_t$ right vertices, and $D_t E_t$ weighted edges where $D_t$ and $E_t$ correspond to total demand and endowment at time $t$ (as given in Definition 2). Since we can upper bound $D_t \leq N$, $E_t \leq M$, and $\min{D_t, E_t} \leq L$, the results in the previous paragraph follow. \
> **Reference:** Ramshaw, L., & Tarjan, R. E. (2012). On Minimum-Cost Assignments in Unbalanced Bipartite Graphs.
>
> * We recognize the importance of the practical considerations you've emphasized, such as privacy concerns, fairness, and strategic user behavior, within real-world applications. While these aspects extend beyond the paper's present focus, we intend to highlight their significance in our revised manuscript. There is an opportunity to delve into these areas more extensively in subsequent research, building on the groundwork we've established. \
> To further address the inquiry regarding strategic user behavior, we wish to offer the following insight: If each user interacts with the system only once, it would be in their best interest to behave myopically. In particular, our frameworks of fixed valuations and random valuations can be used to model settings where each user interacts with the system only once (at a single time step) if each user is associated with a type that determines their valuations. In this case, the set $\mathcal{N}$ corresponds to the set of all user types and each demand parameter $d^t_u$ represents the total demand of users of type $u$ in round $t$. Under the fixed valuations model, all users of type $u$ have valuation $v_{ui}^t = v_{ui}$ for item $i$ at all rounds $t$. Under the random valuations model, each user of type $u$ has a random valuation with distribution $F_{ui}$ for item $i$ independently for each user at each time $t$. Since at most one user receives any item $i$ in any round, it is sufficient to consider a single random valuation $v_{ui}^t$ for each type $u$ and item $i$ at time $t$. Crucially, since each user interacts with the system only once, it is in each user’s best interest to behave myopically.

---

> > ### Comment · Reviewer_ABaL · 2023-08-11
> >
> > Thank you for your response. I have thoroughly reviewed your response as well as the comments provided by other reviewers. After careful consideration, I have decided to maintain my current score.

---

### Author Rebuttal · Authors · 2023-08-09

We thank all the reviewers for their constructive feedback and thoughtful questions. Please see our separate responses below each review.

In response to a question from reviewer Hkdu regarding the experiments, we provide the new results in the PDF file. In our initial submission, Figure 4 was generated for experiments with $N = 30$ users and $M = 20$ items. Now, we have improved the time complexity of our simulations using a maximum weight matching algorithm instead of an LP solver as suggested by reviewers 2rbX and ABaL. Therefore, we are now able to run the experiments in a shorter time and generate results for larger experiments. In our revised manuscript, we are committed to repeating this experiment across multiple runs and replacing Figure 4 with those results.

---

### Decision · Program_Chairs · 2023-09-21

**Decision:**

Accept (poster)

**Comment:**

This paper presents an online pricing framework for multi-user multi-item markets. Online algorithms to efficiently offer and price items while learning user valuations from accept/reject feedback are proposed and analyzed. We hope the reviews and feedbacks during the discussions could be helpful for the authors to prepare the final version of this paper.